# *Mms19* promotes spindle microtubule assembly in *Drosophila* neural stem cells

Rohan Chippalkatti[1,2], Boris Egger[3], Beat Suter[1]*

1 Cell Biology, University of Bern, Berne, Switzerland, 2 Graduate School for Cellular and Biomedical Sciences, University of Bern, Berne, Switzerland, 3 Department of Biology, University of Fribourg, Fribourg, Switzerland

* Beat.Suter@izb.unibe.ch

**Data Availability Statement:** All relevant data are within the manuscript and its Supporting Information files.

**Funding:** This work was supported by funding from the Swiss National Science Foundation

## Abstract

Mitotic divisions depend on the timely assembly and proper orientation of the mitotic spindle. Malfunctioning of these processes can considerably delay mitosis, thereby compromising tissue growth and homeostasis, and leading to chromosomal instability. Loss of functional *Mms19* drastically affects the growth and development of mitotic tissues in *Drosophila* larvae and we now demonstrate that *Mms19* is an important factor that promotes spindle and astral microtubule (MT) growth, and MT stability and bundling. *Mms19* function is needed for the coordination of mitotic events and for the rapid progression through mitosis that is characteristic of neural stem cells. Surprisingly, *Mms19* performs its mitotic activities through two different pathways. By stimulating the mitotic kinase cascade, it triggers the localization of the MT regulatory complex TACC/Msps (Transforming Acidic Coiled Coil/Minispindles, the homolog of human ch-TOG) to the centrosome. This activity of Mms19 can be rescued by stimulating the mitotic kinase cascade. However, other aspects of the *Mms19* phenotypes cannot be rescued in this way, pointing to an additional mechanism of *Mms19* action. We provide evidence that Mms19 binds directly to MTs and that this stimulates MT stability and bundling.

## Author summary

Mitosis is a fundamental process that segregates replicated chromosomes into daughter cells, allowing organ growth and development in multicellular organisms. To properly distribute the genetic material, the mitotic spindle, an organelle consisting of extended microtubules, microtubule motors, and additional microtubule-associated proteins needs to be built in a coordinated, robust, but still dynamic way. Failure to set up these spindles properly leads to chromosomal instability or differentiation defects, and this can lead to tumor formation, reduced organ growth, or lack of specific cell types. Whereas Mms19 protein performs activities unrelated to mitosis, we found that *Drosophila Mms19* is also crucial for mitotic progression and organ growth. This led us to discover that Mms19 had been repurposed to also assist in the formation of stable spindle microtubules. By regulating spindle architecture, *Mms19* allows neural stem cells to timely progress through mitosis to build the normal brain. Surprisingly, *Mms19* exerts its spindle regulatory function

(project grant 31003A_173188; www.snf.ch) and the University of Bern (www.unibe.ch) to BS. The funders had no role in study design, data collection and analysis, decision to publish, or preparation of the manuscript.

**Competing interests:** The authors have declared that no competing interests exist.

again through different activities. It stimulates microtubule assembly through a mitotic kinase cascade consisting of 3 kinases to activate microtubule organizer proteins. Additional evidence suggests that it is capable of interacting with microtubules and promotes microtubule bundling and that this is also important to form a functional mitotic spindle.

## Introduction

The fidelity of chromosome segregation during mitotic divisions depends on the proper regulation of the structure and dynamics of spindle microtubules. Numerous mitotic factors, including different mitotic kinases and microtubule (MT) associated proteins, meticulously regulate the formation, orientation, polymerization, and catastrophe of spindle MTs to ensure that the chromatids are evenly delivered to the two products of the division process, the daughter cells. The mitotic spindle not only regulates chromosome segregation, its proper spindle architecture, and orientation profoundly impacts development and cellular physiology. Spindle size and architecture dynamically adapt to changes in cellular morphology within and between organisms. Early embryonic divisions in organisms such as *C. elegans* and *X. laevis* occur in the large volumes of the egg and blastomeres whereas in the later stages of embryogenesis cell sizes are dramatically smaller [1,2]. The mitotic spindle is scaled according to the cell size by MT-regulatory proteins, indicating that cell division is in tune with the developmental progression of these organisms [3]. The MT associated protein Tpx2 mediates spindle architecture variations in mouse neuronal stem cells (NSCs), where astral MTs predominate the early developmental stages and inner spindle MTs predominate during the later developmental stages [4]. This points to the importance of MT regulatory proteins in regulating spindle architecture during development. Studying such novel proteins can reveal structural and dynamic aspects of spindle assembly and their role in development and cellular physiology.

*Mms19* was identified as a gene required for nucleotide excision repair (NER) [5]. The protein Mms19 can be found as part of the Cytoplasmic Iron-Sulfur Assembly complex (CIA), which mediates the incorporation of iron-sulfur clusters into NER proteins such as Xpd [6,7]. However, a distinct, apparently DNA repair-independent, function of *Mms19* came to light because its knock-down caused numerous mitotic spindle abnormalities in human cells [8]. Downregulation of *Mms19* in young *Drosophila* embryos also revealed spindle abnormalities and chromosome segregation defects, and these phenotypes were linked to the mitotic control pathway when results by Nag and co-workers revealed that Mms19 acts as a positive regulator of the Cdk Activating Kinase (CAK) activity [9]. CAK has dual roles during the cell cycle. It activates the mitotic Cdk1 during mitosis but is recruited by Xpd to form the holoTFIIH complex during interphase [10]. The incorporation into TFIIH causes CAK to phosphorylate an entirely different set of substrates and allows CAK to perform its functions in transcription. Nag et al proposed that Mms19 binds to Xpd during mitosis and that this binding competes with the binding of Xpd to the TFIIH subunits. Mms19 binding to Xpd could thereby release CAK to activate its mitotic targets. In this model, reduced levels of Mms19 prevent sufficient dissociation of the Xpd-CAK complexes, hindering the establishment of the required levels of the mitotic CAK activity. Indeed, overexpressing CAK complex components in the *Mms19* loss-of-function (*Mms19$^P$*) background rescued the mitotic defects to a large degree [9].

The remarkable findings by Nag et al uncovered a novel mitotic pathway for *Mms19*. But this study mostly focused on young *Drosophila* embryos, which are unusual in that all somatic nuclei share a common cytoplasm, in which the mitotic divisions take place. Furthermore, their cell cycle consists of only S and M phases, without intervening G phases. In this situation

with the shared cytoplasm, the absence of Mms19 often causes microtubules emanating from one spindle pole to contact the chromosomes of a neighboring nucleus. It is therefore difficult to extrapolate these findings to mononuclear, diploid cells that are isolated from their neighbors by plasma membranes and go through a full cell cycle with G phases. Furthermore, even though the spindle abnormalities observed in the absence of Mms19 could be linked to compromised CAK activity, the pathway acting downstream of CAK is not understood. Finally, overexpression of the CAK complex brought about only a partial rescue of the *Mms19$^P$* defects, pointing towards additional, possibly CAK-independent spindle regulatory roles of *Mms19*.

The objective of this study was thus to investigate the mitotic function of *Mms19* in normal diploid cells with the goal of dissecting the precise pathway through which *Mms19* acts to regulate mitotic spindle assembly and cell cycle progression. Because *Mms19$^P$* larvae lack imaginal discs, we chose the larval brain neuroblasts (NBs) as a model to analyze the mitotic roles of *Mms19* in cells with a full cycle. The newly identified *Mms19* phenotypes allowed us to pinpoint steps in the pathways that require *Mms19* activity. We found that *Mms19* is required in NBs for timely progression through the cell cycle and consequently for establishing normal cell numbers in the NB lineage. *Mms19* is also required for the growth and assembly of spindle and astral microtubules (MTs). Our results connect these defects to the mis-localization of the microtubule regulator TACC, suggesting that TACC is a crucial downstream target of the mitotic kinase cascade through CAK-Cdk1-Aurora kinases. Additionally, and apparently unrelated to the CAK-Cdk1 axis, we also identified a direct interaction *in vitro* between Mms19 and microtubules and found that Mms19 promotes microtubule stability and bundle formation.

## Results

### *Mms19$^P$* brains show a microcephaly phenotype

Normal *Drosophil*a larvae spend on average 5–7 days at 25˚C to pass through the larval stages. In contrast, *Mms19$^P$* larvae take not only at least 8–10 days to reach the size of outgrown WT third instar larvae, but they also spend a total of around 15 days in the 3$^{rd}$ larval instar stage before they die (see Table 1 for details about the fly stocks used). These larvae display a typical mitotic phenotype without recognizable imaginal disc tissues [9]. Even though outgrown *Mms19$^P$* larvae lack imaginal discs, their brain is still present, although it is much smaller, displaying a microcephaly phenotype (Fig 1A–1B). Additionally, the optic lobe (OL) appears deformed and underdeveloped. Compared to the wild type, the total volume of the *Mms19$^P$* brains and the volumes of the central brains (CBs) and OLs, too, are drastically reduced (Fig 1D–1F). We also stained the brains of all genotypes with antibodies against Miranda (Mira, which marks CB NBs; see Table 2 for details about all antibodies used for staining), to count the number of CB NBs. Surprisingly, however, the number of CB NBs per brain lobe did not significantly change between the controls and the *Mms19$^P$* brains (Fig 1C). This implies that the *Mms19$^P$* NBs have been properly determined and are present. It was reported previously that aneuploidy resulting from defective spindle checkpoint function causes premature differentiation of NBs and a reduction in brain size [11]. We performed fluorescent *in situ* hybridization to screen for aneuploid NBs in the WT as well as *Mms19$^P$* brains but did not find significantly elevated levels of aneuploidy in the mutant (S1 Fig; *P* = 0.1493). The reduced central brain size might then result from dying differentiating neuronal cells or, more likely, from NBs that divide only slowly, thereby contributing fewer Ganglion Mother Cells (GMCs) and neurons to the CB. The OL develops from the invagination of a group of ectodermal cells in the head region during late embryogenesis [12]. In the wild type, these cells, called neuroepithelial (NE) cells, start proliferating after larval hatching and generate the distinct regions of

**Table 1. Fly stocks used for these studies.**

| Fly stocks | | |
| --- | --- | --- |
| Reagent | Description | Source |
| P{EPgy2}Mms19EY00797/TM3, Sb1 Ser1 (Mms19$^P$) | P-element insertion in the third exon of *Mms19* | Bloomington stock center #15477 |
| +; Mms19::eGFP, Mms19$^P$ | eGFP-tagged Mms19 protein expressed in *Mms19$^P$* background | own work [9] |
| +; EB1::GFP/CyO | GFP-tagged EB1 protein expressed under the poly-ubiquitin promoter | provided by Regis Giet |
| +;EB1::GFP/CyO; Mms19$^P$/Tm6,Tb | EB1::GFP expressed in the *Mms19$^P$* background | generated in this study |
| hs-flp/hs-flp; tub-Gal4, UAS-mCD8::GFP/CyO, actin::GFP; FRT82B, tub-Gal80/TM6, Tb | MARCM driver stock | provided by Bruno Bello |
| +; FRT82B, Mms19$^P$/TM6, Tb | *FRT82B* recombined with *Mms19$^P$* | own work [9] |
| da-Gal4/CyO, mCherry; UAST-Mat1, Mms19$^P$/TM3, Ser, GFP | Stock for overexpression of CAK in the *Mms19$^P$* background | own work [9] |
| UAST-Cdk7/CyO, mCherry/ UAST-CyclinH, Mms19$^P$/TM3, Ser, GFP | Stock for overexpression of CAK in the *Mms19$^P$* background | own work [9] |
| UAST-Cdk7/CyO, mCherry/ UAST-CyclinH /TM3, Ser, GFP | Stock for overexpression of CAK in the wild-type background | own work [9] |
| da-Gal4/CyO, mCherry; UAST-Mat1 /TM3, Ser, GFP | Stock for overexpression of CAK in the wild-type background | own work [9] |

the OL after several rounds of divisions [13]. The drastic reduction of the OL volume seen in *Mms19$^P$* larvae contributes the largest part to the overall reduction of the brain size seen. Although this points to a requirement for *Mms19* also in these highly proliferative NE cells, we focused on the better studied CB NBs as these cells are large and very well suited to analyze mitotic phases and possible defects in spindle assembly.

To verify the requirement for Mms19 for normal brain development, we expressed wild-type Mms19 fused to eGFP (Mms19::eGFP) under the control of the endogenous *Mms19* promoter in the *Mms19$^P$* background to test whether the observed phenotype can be rescued by *Mms19* activity. Mms19::eGFP fully rescued the brain morphology, the volumes of the whole brain lobe as well as the CB and OL sizes (Fig 1A, 1D–1F). This not only confirmed that the defects observed in the mutant are indeed due to the absence of *Mms19* activity and not due to a second site mutation on this chromosome, it also showed that the Mms19::eGFP fusion protein is functional.

A mitotic activity of *Mms19* was described to promote the Cdk-activating kinase activity of the Cdk7/CycH/Mat1 complex (CAK complex; [9]). The responsible mechanism appears to be a competitive binding of Mms19 to Xpd, which would otherwise recruit CAK to TFIIH, where it assumes a different substrate specificity and is unable to activate the M-Cdk Cdk1. The key result that led to this model was that the lack of imaginal discs caused by the absence of *Mms19* activity could be rescued considerably by expressing the CAK complex components under the control of the Upstream Activating Sequence (UAS) enhancer, using the *daughterless*(*da*)-Gal4 driver in the *Mms19$^P$* background (da>CAK, *Mms19$^P$*) [9]. We, therefore, tested whether it was possible to rescue the small brain phenotype by the identical strategy. This was not the case, as in da>CAK, *Mms19$^P$* brains both CBs and OLs remained smaller (Fig 1A, 1D–1F), the number of CB NBs was similar as in the wild type (Fig 1C) and the overall brain volume was reduced. These observations indicated that the mitotic cells in the *Mms19$^P$* brains probably did not proliferate enough to produce the normal amount of neuronal tissue and that this defect might not only be caused by insufficient CAK activity. We also considered whether the expression of CAK under *da-Gal4* control might cause the abnormalities leading to the reduced

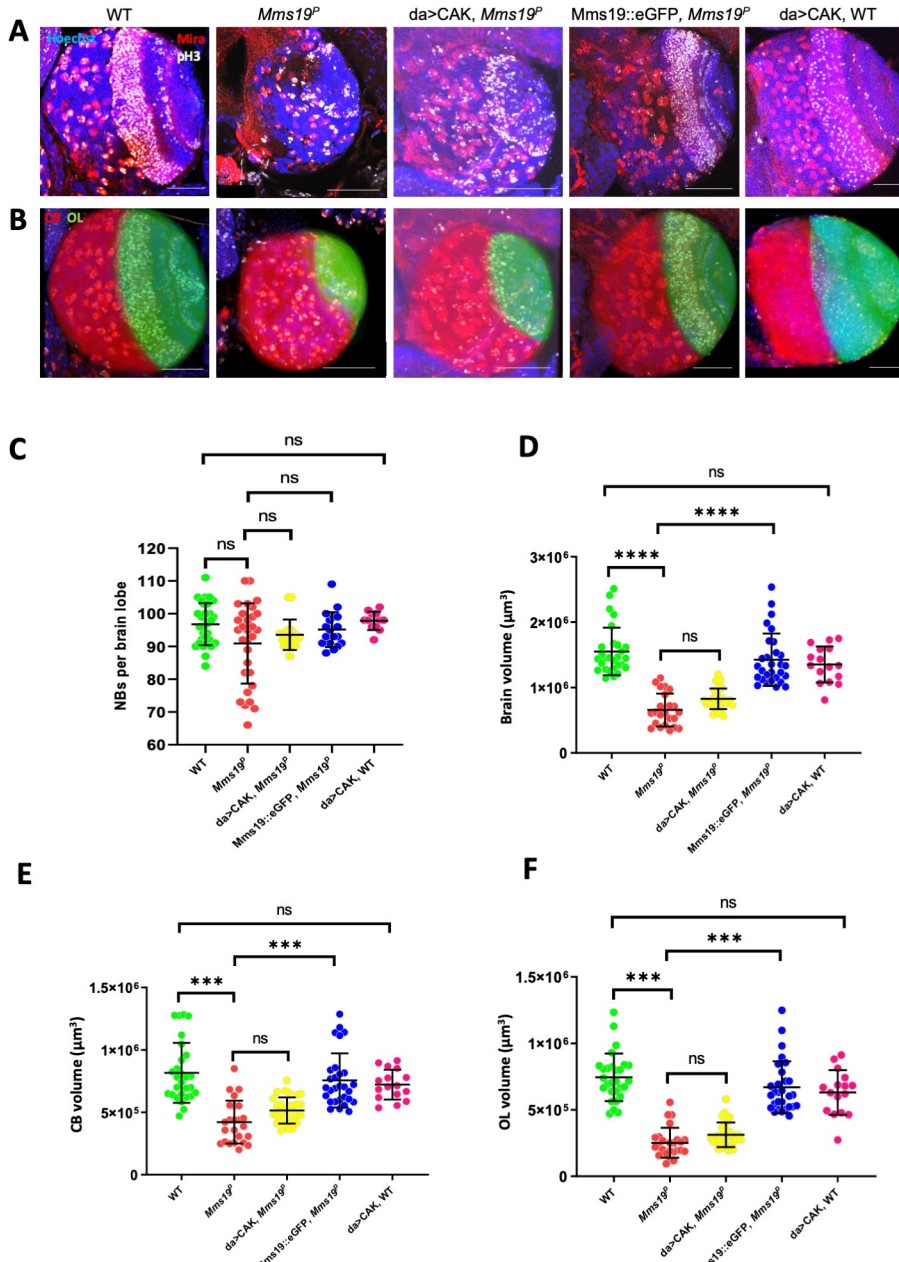

**Fig 1. *Mms19^P* brains display a microcephaly (small brain) phenotype.** 3rd instar larval brain NBs were visualized by staining for Miranda (red, cytoplasmic). They were also stained for pH3 (white, nuclear) and DNA (blue, Hoechst 33342 dye). (A)-(B) A WT brain lobe can be subdivided into the OL (shaded green) and the CB (shaded red). *Mms19^P* brains are significantly smaller, with a diminished OL. Overexpressing the CAK complex components driven by daughterless-Gal4 (da>CAK) in the *Mms19^P* background appeared to slightly rescue this phenotype but this rescue was not statistically significant (brain volume: *P*>0.99; OL volume: *P*>0.99). Mms19::eGFP expressed in the *Mms19^P* background rescued the *Mms19^P* phenotype. Daughterless driven CAK expression in the WT background did not seem to bring about any noticeable defects in brain size or morphology. Counting the number of CB NBs (in the Red shaded area) staining positively for Mira revealed that the number of NBs per brain lobe is similar across all genotypes (C) (WT: n = 15 brains, 30 lobes, 3 experiments; *Mms19^P*: n = 15 brains, 30 lobes, 3 experiments; da>CAK, *Mms19^P*: n = 9 brains, 18 lobes, 3 experiments; Mms19::eGFP, *Mms19^P*: n = 9 brains, 18 lobes, 3 experiments; da>CAK; WT: n = 6 brains, 12 lobes, 2 experiments). (D) the volume of the brain is significantly reduced in the *Mms19^P* and da>CAK, *Mms19^P* brains. Furthermore, segmentation and volume measurement of the OL and CB revealed a significant reduction in *Mms19^P* and da>CAK, *Mms19^P* brains (E)-(F). Brain volume and morphology were restored to WT levels when Mms19::eGFP was expressed in *Mms19^P* brains (D)-(F). (WT: n = 14 brains, 28 lobes, 3 experiments; *Mms19^P*: n = 11 brains, 22 lobes, 3 experiments; da>CAK, *Mms19^P*: n = 15 brains, 30 lobes, 3

experiments; Mms19::eGFP, *Mms19$^P$*: n = 14 brains, 28 lobes, 3 experiments; da>CAK; WT: n = 8 brains, 16 lobes, 2 experiments). Statistical significance (SS) was determined by the Kruskal-Wallis test. Multiple columns were compared using Dunn's post test, ****($P<0.0001$), ***($P<0.001$), scale = 50µm.

brain size. For this, we dissected brains from larvae expressing da>CAK in the wild-type background. However, we did not find significant changes in either the brain morphology (Fig 1A), the CB NB numbers (Fig 1C), or the CB and OL volumes (Fig 1E and 1F). This result, therefore, points to the possibility that Mms19 acts through two different pathways to achieve normal organ size.

## In *Mms19$^P$* mutant brains a higher proportion of NBs are in mitosis

In order to better understand the microcephaly phenotype, we performed 5-Ethynyl-2'-deoxyuridine (EdU) incorporation assays coupled with phosphorylated Histone H3 (pH3) staining and examined defects in cell cycle progression in the NBs. Based on pH3 and EdU staining, the cells can be allocated to one of the following phases: only EdU = S, i.e. cells in S phase; pH3 without EdU = M, i.e. cells undergoing mitosis; both EdU and pH3 = G2/M, i.e. cells transiting from S to G2/M phase; and neither EdU nor pH3 (= G1/G0, i.e. Gap phase; S2A Fig). As it is difficult to determine from pH3 staining alone, whether the cells are in the G2 phase or in M, we combined the double-positive cells as well as the pH3 positive cells in the same category and referred to this as 'mitotic phase'. CB NBs were additionally marked with antibodies against Miranda (Mira) and the number of cells was counted in each class. The results are represented as a percentage of the total number of NBs per brain lobe. We observed that about half the NBs were not labeled (i.e. G1/G0 cells: 47–56%; S2B Fig) and the relative differences between the genotypes were small. However, lack of *Mms19* caused a clear increase in the fraction of cells in the mitotic phase (37% compared to 28% in the wild type; S2D Fig). This result could either mean that more cells undergo divisions or that the *Mms19$^P$* NBs are either trapped in M phase or proceed more slowly through it.

**Table 2. Primary and secondary antibodies used for immunostainings.**

| Primary antibodies | | | |
|---|---|---|---|
| **Antibody** | **Manufacturer** | **Catalog number** | **Dilution** |
| Rat anti-Miranda | Abcam | Ab197788 | 1:250 |
| Rabbit anti-alpha Tubulin | Abcam | Ab18251 | 1:500 |
| Mouse anti-alpha Tubulin | Sigma | T6199 | 1:500 |
| Rabbit anti-GFP | Immunokontakt | 210-PS-1GFP | 1:500 |
| Rabbit anti-pH3 | Cell Signaling | 9701 | 1:200 |
| Rabbit anti-Mms19 | Genscript | Custom antibody | 1:500 |
| Mouse anti-γ-Tubulin | Sigma | T6557 | 1:1,000 |
| Rabbit anti-TACC | provided by Jordan Raff | - | 1:1,000 |
| Rabbit anti-Msps | Provided by Hiro Ohkura | - | 1:1,000 |
| Rabbit anti-Aurora A | Provided by Jurgen Knoblich | - | 1:200 |
| **Secondary antibodies** | | | |
| Goat anti-rat Cy3 | Jackson Immuno | 112-165-167 | 1:150 |
| Goat anti-mouse Alexa 488 | Invitrogen | R37120 | 1:500 |
| Goat anti-rabbit Alexa 488 | Invitrogen | A27034 | 1:500 |
| Goat anti-mouse Alexa 647 | Invitrogen | A21235 | 1:500 |

## NBs depend on *Mms19* for timely and coordinated spindle assembly and orientation

To study how *Mms19* contributes to spindle assembly and progression through mitosis, we utilized a transgene that expresses EB1::GFP, a MT plus end binding and tracking protein that labels growing MT ends [14]. Live imaging of NBs expressing EB1::GFP revealed the dynamics of the formation of the mitotic spindle and allowed us to measure progression through the early part of the M-phase, including the period from <u>N</u>uclear <u>E</u>nvelope <u>B</u>reak-<u>D</u>own (NEBD) to the onset of anaphase B. NEBD was determined by the appearance of the GFP signal in the nuclear region, which lacks a GFP signal until NEBD. To determine the onset of anaphase B, we analyzed the distribution of EB1::GFP and the spindle morphology throughout mitosis. EB1 localization and behavior during anaphase B has been studied previously [15–17]. At the beginning of anaphase B, the spindle starts to elongate and this elongation is accompanied by an overall reduction in EB1::GFP fluorescence. We observed this pattern in WT NBs, where at 7 min post NEBD, the spindle slightly elongated, concomitant with a decrease in EB1::GFP fluorescence. NBs expressing EB1::GFP in the wild-type background started anaphase B approximately 6–8 min after NEBD (Fig 2A and 2C; S1 mov). On the other hand, NBs expressing EB1::GFP in an *Mms19*[P] background reached anaphase B onset only around 16–20 min after NEBD (Fig 2B and 2C; S3 mov).

Interestingly, in some of the EB1::GFP expressing *Mms19*[P] NBs, the spindle formation started before the two centrosomes had finished migrating to the opposite sides of the nucleus (Fig 2B; S3 mov). As a result, at 3 min, a kinked spindle was observed, which eventually straightened out at 5 mins. To quantify defects in centrosome migration/positioning, we used a method described previously [18] to measure the angle between the 2 centrosomes just before NEBD (Fig 2D). In all of the WT cells, the angle between the centrosomes was in the range of 135° to 180°. But in around 20% of *Mms19*[P] NBs, severe centrosome mispositioning was noted as the angles fell into the range of 45° to 135° (E). In one case of an *Mms19*[P] NB spindle, only one centrosome started nucleating MTs (S5C Fig; S8 mov). The spindle remained monopolar until 6 min post-NEBD and only then, a bipolar spindle became apparent. Further, around 20% of the spindles in *Mms19*[P] NBs changed their orientation during the course of mitosis (Fig 2B; S3 mov), while all wild-type spindles examined remained firmly anchored at the cortex and did not change their orientation (Fig 2A).

*Drosophila* larval NBs are characterized by a defined apical-basal polarity [19] where the atypical Protein Kinase C (aPKC)-Bazooka-Pins complex localizes to the apical cortex while Miranda localizes to the basal cortex (Fig 2F). Retention of NB identity and self-renewal depends on the inheritance of the aPKC-containing apical complex whereas differentiation of the GMC relies on the inheritance of the Mira-bound cargo of the basal cortex [23]. Therefore, the orientation of the mitotic spindle is tightly coupled to this apical-basal polarity axis [20,21]. We therefore further examined the spindle orientation defects in fixed cells labeled with the polarity marker Miranda (Mira), which localizes to the basal cortex of the NB, forming a crescent-like pattern. We found that most of the wild-type spindles form at an angle of 10 degrees or lower relative to Mira (Fig 2G, 2H and 2J). On the other hand, *Mms19*[P] NBs more frequently failed to align their spindles within 10 degrees (Fig 2G, 2I and 2J) and the largest tilt that we observed was around 60°. However, a marked effect on cell fate determination is only observed when the spindle angle gets close to 90° [22,23]. In such cases, even though both daughter cells inherit a part of the Mira-containing basal cargo, this is not sufficient to drive differentiation. Instead, both cells assume a NB identity, leading to an increase in NB numbers per brain lobe [23]. Consistent with this spindle orientation defect being insufficient to cause major NB amplifications, we did not see a significant difference in NB numbers per brain lobe

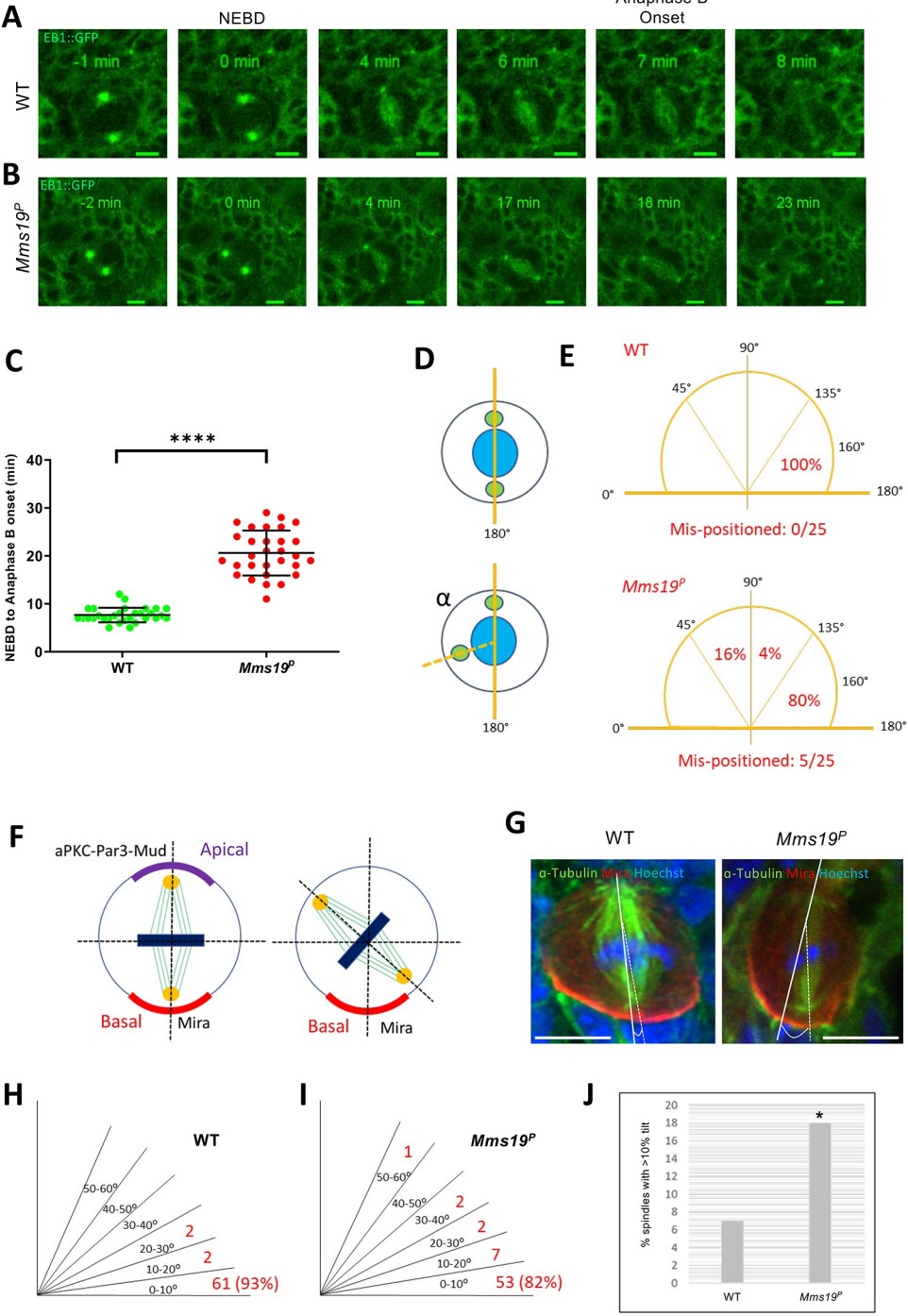

**Fig 2. NBs depend on *Mms19* for timely and coordinated spindle assembly and orientation.** (A-C) WT NBs typically start anaphase B 6–7 min after the onset of NEBD. (B) *Mms19^P* NBs require on average 15–20 min to reach anaphase B onset. (B-C) n = 30 NBs per genotype, 1 experiment. The quantitative assessment of the duration of WT and *Mms19^P* mitoses is compared in (C). SS was determined by an unpaired t-test (***$P < 0.001$); Scale = 5μm. In this particular *Mms19^P* NB (B) the spindle poles are not fully separated at NEBD and the spindle initially appears kinked. (D) To analyze centrosome positioning defects, we measured the angle between the two centrosomes relative to the center of the nucleus, just before NEBD. (E) In most WT NBs, the angle between the centrosomes fell between 135° and 180°. However, 20% of the mutant NBs displayed centrosome separation defects as the angle was lower than 135°. N = 25 per genotype, 1 experiment. Additionally, in some of the *Mms19^P* NBs analyzed, the spindles changed their orientation throughout the course of mitosis, indicating spindle orientation defects. (F) Spindle orientation in the WT

NBs is tightly coupled to the apical-basal polarity axis and this is reflected by the angle formed by the spindle with respect to the basal Mira crescent. (G) In a fraction of *Mms19^P* NBs the spindle appears misoriented with respect to (w. r.t) the Mira crescent localization. (H-J) Quantification of the spindle angle revealed that a considerably higher number of *Mms19^P* NBs are oriented at an angle of more than 10˚ w.r.t Mira. Significance was calculated using Fisher's exact test, *($P = 0.0309$). n = 65 NBs per genotype, 3 experiments, scale = 5µm.

between the wild type and *Mms19^P* (Fig 1C). Therefore, the relatively minor spindle orientation defect due to lack of *Mms19* does not appear to lead to differentiation problems but impedes efficient mitosis. We conclude that in the absence of *Mms19*, spindle formation is not properly coordinated with cell cycle progression. Furthermore, defects in centrosome migration and in spindle assembly and orientation contribute to mitotic delays in *Mms19^P* NBs.

## *Mms19* is cell autonomously required to maintain normal cell numbers

In the experiments with the *Mms19^P* mutants, Mms19 was absent from all larval cells. In this situation, the mitotic delay could be due to a systemic effect or due to the lack of a cell-autonomous activity of Mms19. To test whether *Mms19* is specifically required in the mitotic cells for the timely progression through mitosis, we generated *Mms19^P* mosaic NB clones in an otherwise wild-type background. Mosaic clones were induced in NBs 24hrs after larval hatching (ALH) and the expansion of these clones was analyzed by dissecting the brains in mid-third instar larvae 72hrs ALH. For this experiment, we focused on the type I NBs on the ventral side. On average, 45 cells per clone were found in control clones, but only around 30–35 cells in *Mms19^P* clones ($P<0.05$; S3A–S3C Fig), indicating that the cell-autonomous loss of *Mms19* activity hinders the establishment of normal cell numbers in the NB lineage. This observation reaffirms our conclusion that the absence of *Mms19* delays mitosis in NBs and that this mitotic delay probably causes more *Mms19^P* NBs to linger in M-phase (S2 Fig).

## *Mms19* is required to form spindles of normal length and density

*Mms19^P* spindles were found to be generally shorter than the wild-type ones (Fig 3A–3D). In normal NB spindles, the centrosomes were anchored close to the cell cortex, with spindle MTs emanating from them and extending to the chromosomes (Fig 3A and 3A'). But in around a quarter of the mutant cells, even though the chromosomes were aligned at the metaphase plate, the centrosomes were not connected to the cell cortex and were only a short distance away from the metaphase plate (Fig 3B and 3B'). In order to quantify this defect, we measured the length of the spindles across all genotypes and found that the spindles in *Mms19^P* and da>CAK, *Mms19^P* were generally shorter than the wild-type control spindles (S3 Table). As the NBs were also considerably smaller in *Mms19^P* and da>CAK, *Mms19^P* brains, we additionally displayed the spindle length relative to the cell diameter (Fig 3C). A ratio closer to 1 indicates that the centrosomes were anchored close to the cell cortex, as in a healthy spindle. On the other hand, if the ratio was equal to or lower than 0.6, the spindle was defined as a 'short spindle' because the centrosomes were further inside the cell. We found a significant reduction in this ratio in *Mms19^P* and da>CAK, *Mms19^P* NB spindles (Fig 3C and 3D). *Mms19^P* NBs contained around 20% short spindles while this number went up to 30% for da>CAK, *Mms19^P* (Fig 3D). The failure of CAK to rescue the *Mms19^P* phenotype was not due to mitotic defects induced by over-active CAK; because NBs overexpressing CAK in the wild-type background displayed a spindle length/ cell diameter ratio comparable to wild-type NBs (Fig 3C and 3D). On the other hand, expression of Mms19::eGFP in the *Mms19^P* background rescued the short spindle phenotype and only 3% of the NBs displayed it. These findings once

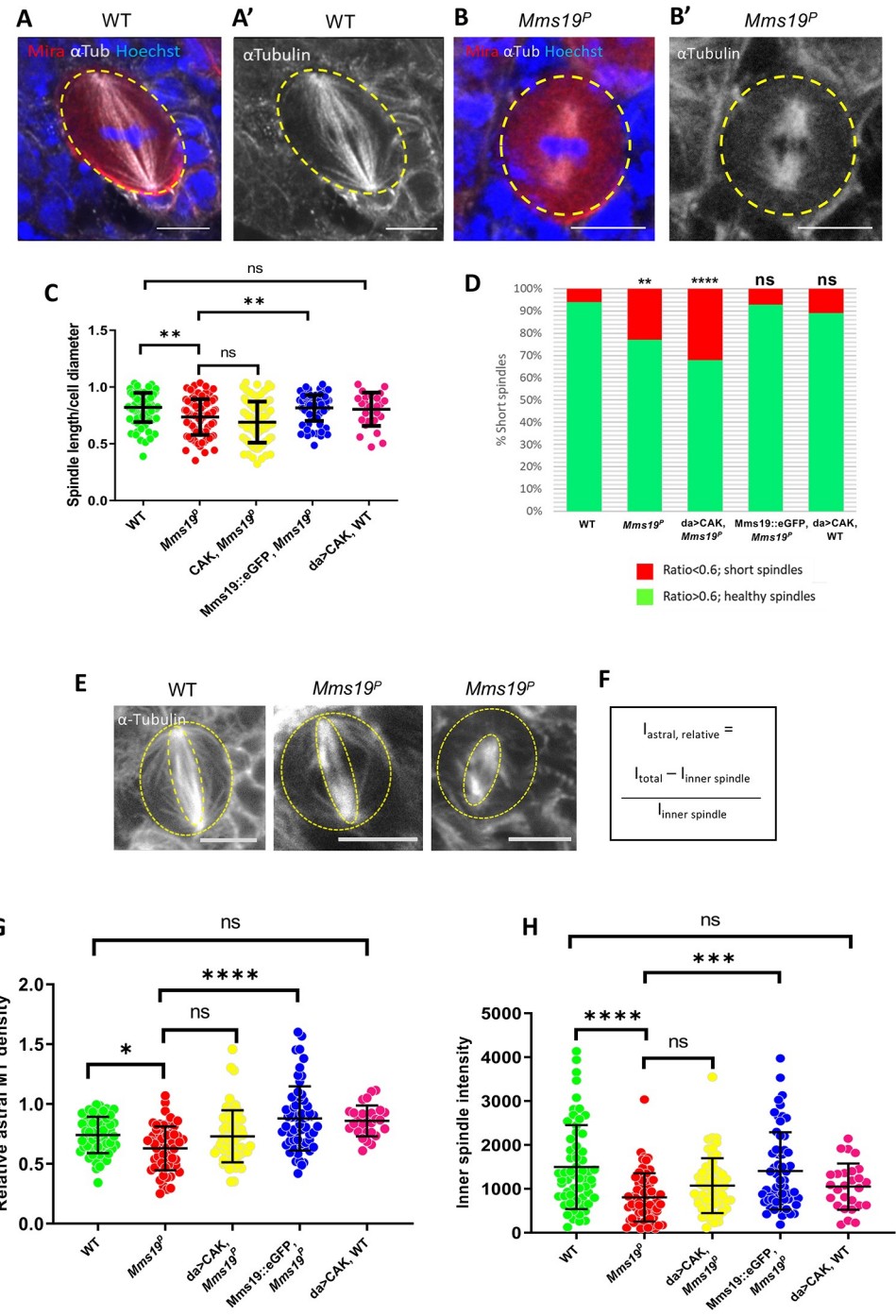

**Fig 3. *Mms19* is required to form a spindle of normal length and density.** (A-A') Z projection of a typical bipolar spindle in a wild-type (WT) NB with the spindle poles anchored to the cell cortex. (B-B') short spindle found in *Mms19^P^* NBs. The spindle is abnormally short and the spindle poles are detached from the cortex. (C) In order to quantify spindle elaboration, we normalized the spindle length to the cell diameter and calculated the ratio. This ratio decreases significantly for *Mms19^P^* NBs. (\*\**P* = 0.0027). This phenotype is rescued by expressing Mms19::eGFP in the *Mms19^P^* background (\*\**P* = 0.0067) but da>CAK fails to rescue the short spindles (*P*>0.99). SS was calculated using Kruskal-Wallis test, columns were compared using Dunn's post test. (D) If this ratio was equal to or less than 0.6, we considered the spindle as 'short spindle'. The graph compares the percentage of 'short spindles' across different genotypes. Significance was calculated using Fisher's exact test, (\*\*\*\**P*<0.0001, \*\**P* = 0.0025). WT; *Mms19^P^*; da>CAK, *Mms19^P^*; Mms19::eGFP, *Mms19^P^*: n = 90 NBs, 3 experiments. da>CAK, WT: n = 30, 2 experiments. (E-F) Relative density of astral MTs was quantified and compared across all genotypes. Maximum intensity projections of mitotic

NBs were obtained (E) and the relative density of astral MTs was quantified by first calculating the inner spindle MT density (inner dotted ellipse) and then subtracting this from the MT intensity from the entire cell (outer dotted circle). This value was then divided by the inner spindle intensity to obtain the relative astral MT density (F). (G) Astral MT density and (H) inner spindle fluorescent intensity was measured in fixed NBs immunostained for α-Tubulin and compared across all the genotypes. Both astral and inner spindle densities were decreased significantly in *Mms19*[P] NBs as compared to WT NBs. Additional expression of da>CAK appeared to partially rescue the phenotype, but this rescue was not statistically significant (astral: $P$ = 0.2633; inner spindle: $P$ = 0.1789). Mms19::eGFP, when expressed in the *Mms19*[P] background, rescued the phenotypes. WT; *Mms19*[P]; da>CAK, *Mms19*[P]; Mms19::eGFP, *Mms19*[P]: n = 60 NBs, 3 experiments. da>CAK, WT: n = 27, 2 experiments. SS was calculated using Kruskal-Wallis test, columns were compared using Dunn's post test. ****$P$<0.0001, ***$P$<0.001; **$P$<0.01, *$P$<0.05, scale = 5μm.

again highlight the possibility of a CAK independent function of Mms19 in regulating spindle architecture.

To determine the effect of *Mms19* on MT formation and stability, we measured astral as well as inner spindle MT density. For astral MT quantification (Fig 3F), we used the method described by Yang and co-workers [24] and found a significant reduction in the *Mms19*[P] NBs as compared to the wild type (Fig 3E–3G). The inner spindle density was also reduced in the *Mms19*[P] NBs (Fig 3H). This phenotype appeared to be slightly rescued by CAK overexpression and was fully rescued by expressing Mms19::eGFP in the *Mms19*[P] background. Reduced astral MT stability could be linked to the spindle positioning and orientation defects described previously (Fig 2B, 2E, 2G–2J) as astral MTs were shown to contact the cell cortex and regulate spindle positioning [25]. *Mms19* is thus necessary for the formation of fully assembled spindles and astral MTs.

## *Mms19* assists MT polymerization *in vivo*

To assess whether the abnormal spindle phenotypes in *Mms19*[P] NBs could arise due to defects in MT growth *in vivo*, we studied NBs expressing EB1::GFP. Live imaging of EB1::GFP revealed the path of elongation of single MTs, akin to that of a comet, and the movement speed of the GFP signal reflects the growth speed of the MT plus ends. Wild-type and *Mms19*[P] larval brains expressing EB1::GFP were dissected and live imaging was performed. The speed of the EB1::GFP particles was then tracked manually using ImageJ. The measurements indicated that the spindle MTs of wild-type NBs polymerized at a rate that is significantly higher than the rate observed in *Mms19*[P] NBs (Fig 4A–4C; WT- S4 mov; *Mms19*[P]- S5 mov). To assess whether the MT assembly function of *Mms19* is restricted to mitotic spindles, we also tested whether the absence of *Mms19* also affects MT polymerization in the post-mitotic glia cells. Measuring EB1 comet speeds in glia cells showed a decrease in MT growth for *Mms19*[P] glia as compared to wild-type glia (Fig 4D–4F; WT- S6 mov; *Mms19*[P]- S7 mov). These results established that *Mms19* assists MT polymerization and growth *in vivo* in mitotic NBs and in post-mitotic glia. The reduced MT growth rate described here seems to be a major factor for the delay in spindle assembly observed in *Mms19*[P] brain NBs. Whereas approximately 3 minutes after the onset of NEBD, wild-type spindles had fully assembled (Fig 2A; S1 mov), most *Mms19*[P] 'spindles' appeared to be short at 3min post-NEBD, and their MT density seemed much lower than in the wild type (S2 mov; S8 mov).

## *Mms19* is required for spindle re-assembly

To learn more about the *Mms19* function in spindle MT growth, we performed the *in vivo* spindle re-growth assay described by Gallaud and co-workers [18]. With this procedure, the NB spindles were depolymerized by incubation on ice for 30 min (Fig 5A, left panel). Large centrosomal asters were observed in the WT NBs after they were shifted back to 25°C for 30s

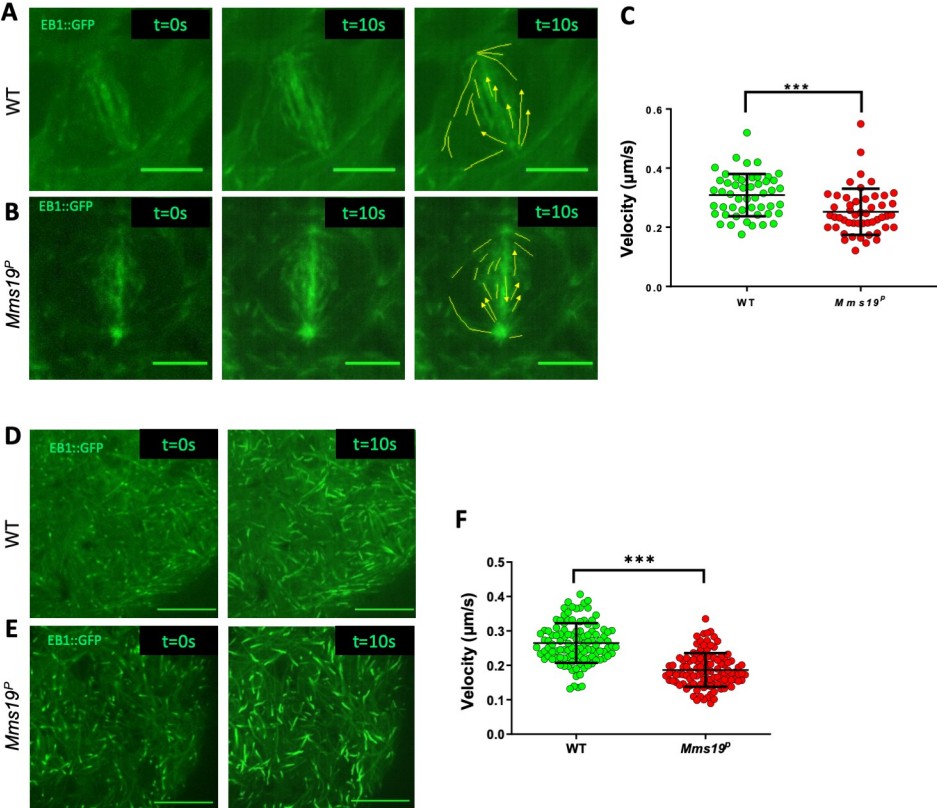

**Fig 4. *Mms19* assists MT polymerization *in vivo*.** (A-B) WT and *Mms19*[P] NBs expressing EB1::GFP were imaged live, and time-lapse movies were acquired to calculate the velocity of EB1::GFP labelled MT 'plus' ends. The left panel shows the image of a single time frame. The central panel shows a projection of 20 time points (taken over 10s). In the far right panel, individual EB1::GFP comets are tracked. The velocities represented in (C) show reduced MT growth velocity in *Mms19*[P] NB spindles. n = 11 cells per genotype, 4–5 tips analyzed from each cell, 1 experiment. (D-E) EB1:: GFP expressing surface glia cells were imaged live to analyze the MT plus end velocities. The left panel shows a single time frame and the right-side panel shows a projection of 20 time points (taken over 10s). The velocities represented in (F) show reduced MT growth velocity in *Mms19*[P] glia. n = 15 brains from each genotype, 7–8 tips analyzed from each brain, 1 experiment. SS for C and F was calculated using unpaired t-test, (***$P < 0.001$), scale bar = 5μm.

(Fig 5A, central panel). MT fibers also appeared to be nucleated around the chromatin (i.e. in the central region between the centrosomes; Fig 5A, central panel). Wild-type NB spindles then regained their standard size and morphology within 90 sec after being shifted back to 25°C (Fig 5A, right panel). In *Mms19*[P] NBs, we also observed a weak astral MT mesh at 30s. However, the spindles failed to re-form to the normal shape after 90 sec. Instead, they remained abnormally short (Fig 5B and 5F). Interestingly, whereas the Mms19::eGFP fusion protein was able to rescue this phenotype, CAK overexpression was unable to do so (Fig 5C and 5D). To validate this short spindle phenotype, we calculated the spindle sizes relative to the cell diameter after 90 sec incubation at 25°C (Fig 5F). Even though *Mms19*[P] cells are smaller, their normalized spindle size was still significantly smaller than the wild-type one. When CAK was overexpressed in *Mms19*[P] NBs, the normalized spindle length did not show a significant rescue even though Mms19::eGFP was able to rescue the *Mms19*[P] mutant phenotype (Fig 5F and 5G). The lack of rescue by CAK overexpression does not seem to be caused by a CAK gain-of-function activity, because, again, wild-type NBs that overexpressed CAK did not display such defects in spindle length (Fig 5E–5G). This result shows that *Mms19* has an

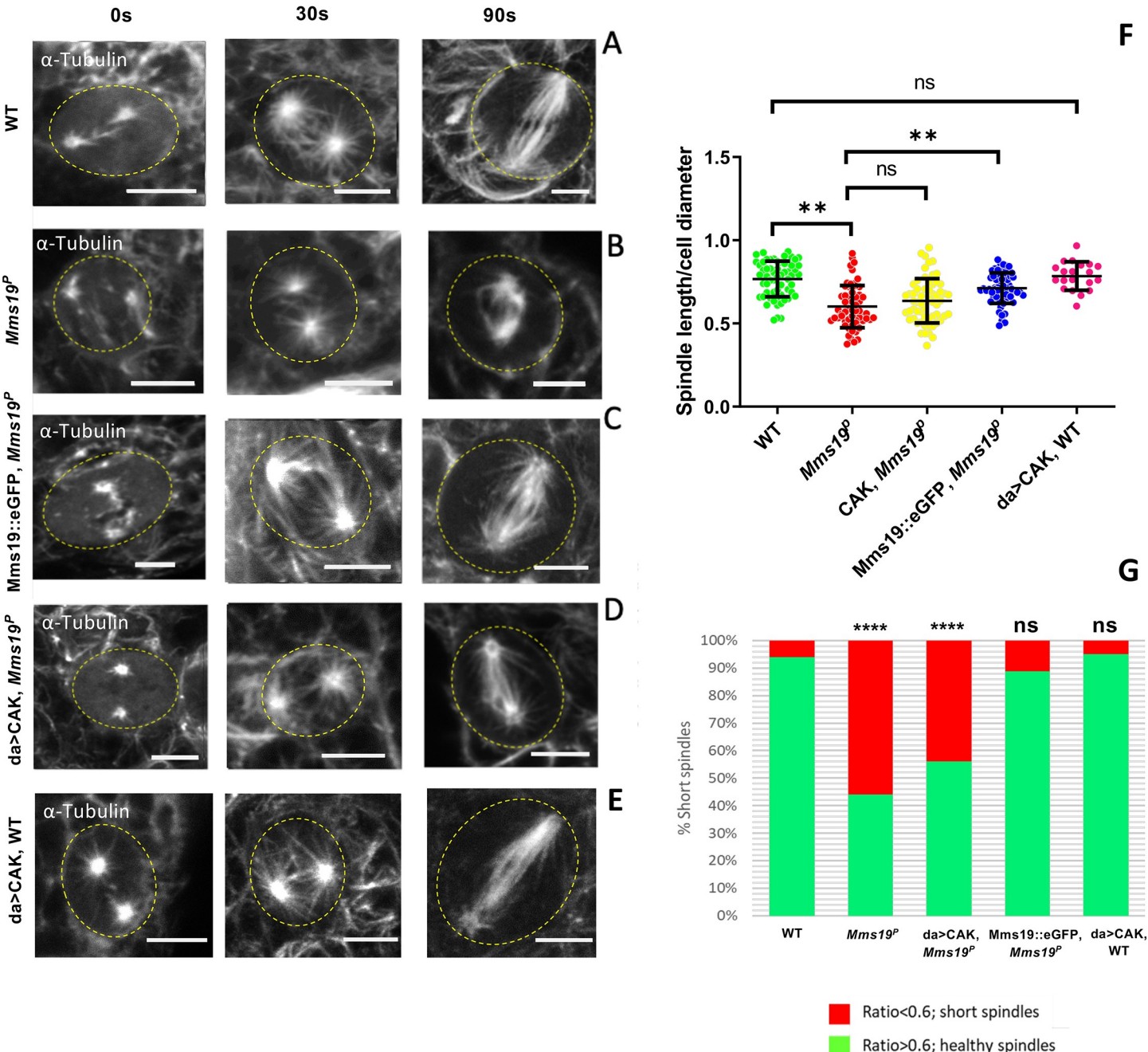

**Fig 5. *Mms19* is required for spindle re-assembly.** After cold treatment, spindle re-assembly was analyzed at 0 sec (i.e. immediately after cold treatment; left panel), 30 sec (central panel), and 90 sec (right panel) after shifting them to 25˚C. For this, the tissue was fixed and immunostained for α-tubulin. At 0 sec in the WT NBs (A), only centrosomes were visible, but after incubation at 30 sec, few fibers had nucleated from the centrosomes and around the chromatin. At 90 sec, the spindle regained its normal shape and density. In *Mms19^P* NBs (B), spindles did not regain the normal shape after 90 sec. Additionally, the microtubule density was reduced in these stunted spindles. Expression of Mms19::eGFP in the mutant background (C), rescued spindle reformation, but overexpression of CAK (D) rescued only slightly and the length and density of MT still appeared reduced (B,D,F: *P*>0.99). (E) Overexpression of CAK in the WT background did not affect spindle re-growth. (F) The scatter plot shows the spindle length relative to the cell diameter after 90 sec incubation at 25˚C. The normalization eliminates variations due to varying cell sizes. SS was calculated by Kruskal-Wallis test, columns were compared by Dunn's post test (***$P$<0.001), (**$P$<0.01). (G) The graph compares the percentage of 'short spindles' across different genotypes. SS was calculated using Fisher's exact test (****$P$<0.001). Scale = 5μm. WT; *Mms19^P*; da>CAK, *Mms19^P*; Mms19::eGFP, *Mms19^P*: n = 60 NBs, 3 experiments. da>CAK, WT: n = 20, 2 experiments.

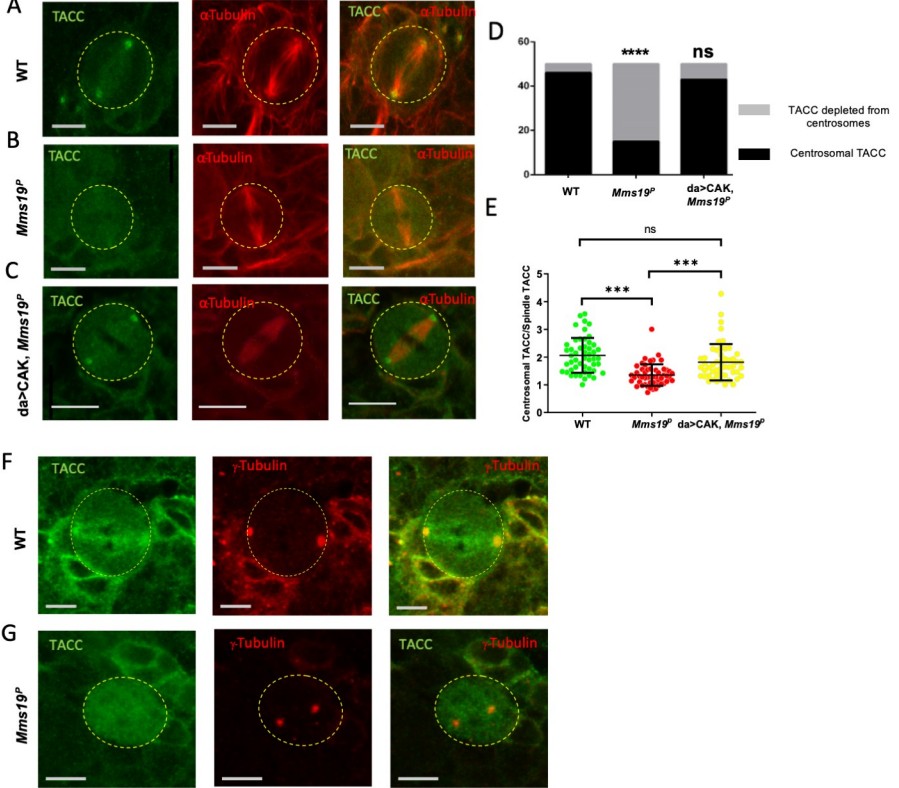

**Fig 6. Centrosomal localization of the MT regulator TACC depends on *Mms19*.** (A) A mitotic NB shows TACC localization at the spindle poles. (B, D) >50% of analyzed *Mms19*[P] NBs fail to localize TACC to spindle poles. SS was calculated using Fisher's exact test (****$P<0.0001$). (C, D) TACC localization was restored upon expression of additional CAK subunits (Cdk7, CycH, and Mat1) in the *Mms19*[P] background. (E) The amount of TACC localized on the centrosomes was quantified by comparing the fluorescent intensity of the centrosomal TACC signal to the TACC signal on the spindles. A ratio greater than 1 was considered as clear centrosomal TACC, while a ratio equal to or less than 1 indicated unlocalized TACC. SS was calculated by Kruskal-Wallis test, columns were compared by Dunn's post test (***$P<0.001$), scale = 5μm. n = 25 NBs per genotype, 3 experiments. (F, G) NBs were stained with antibodies against γ-Tubulin to test for centrosomal localization. (F) TACC co-colocalized with γ-Tubulin on wild-type centrosomes. (G) In the mutant NB, TACC fails to concentrate at γ-Tubulin foci.

important role in establishing proper spindle MT length and that a CAK independent activity of Mms19 is also involved in this.

## Centrosomal localization of the MT regulator TACC depends on *Mms19*

Transforming Acidic Coiled-Coil (TACC), a downstream target of Aurora A kinase, is a critical regulator of centrosomal MTs. During mitosis, Aurora A phosphorylates TACC and stimulates its localization on the centrosomes, where TACC further recruits mini-spindles (Msps). Loss-of-function mutations in either *Aurora A*, *TACC*, or *msps* show drastic abnormalities in astral and spindle MTs [26,27]. As *Mms19*[P] NBs display defects in centrosomal separation, spindle length, and astral MTs, we examined whether Aurora A and TACC might be involved in the same process as Mms19 and if the absence of functional *Mms19* impedes TACC localization and function. We, therefore, stained wild-type and *Mms19*[P] NBs for TACC and observed a strong signal at the wild-type centrosomes (Fig 6A). On the other hand, in >50% of *Mms19*[P] NBs, TACC did not show any enrichment on the centrosomes (Fig 6B and 6D). Similar results were also obtained with the TACC interactor Msps (S4A–S4D Fig). As TACC acts downstream

of the Aurora A kinase, we also examined the localization of Aurora A and found it to be depleted from centrosomes in $Mms19^P$ NBs (S4E–S4G Fig). Because Aurora A itself acts downstream of Cdk1 [28], this defect might be caused by insufficient CAK activity in *Mms19* mutants [9]. We tested this hypothesis by over-expressing the three CAK components in the $Mms19^P$ background (da>CAK, $Mms19^P$). Indeed, upon CAK overexpression, the fraction of spindles displaying centrosomal TACC almost reached wild-type levels (Fig 6C and 6D). To quantify the enrichment of TACC on centrosomes, we compared the fluorescence intensity of centrosomal TACC to the TACC fluorescence intensity on the spindle. This ratio decreased in $Mms19^P$ NBs but was rescued in da>CAK, $Mms19^P$ NBs (Fig 6E). It was also reported that *Aurora A* loss-of-function can cause centrosome fragmentation [26]. Co-staining $Mms19^P$ NBs with antibodies against the centrosomal protein γ-Tubulin showed that even when the TACC signal was not seen on centrosomes, the centrosomal γ-Tubulin was still present, indicating that TACC mislocalization is not caused by centrosome fragmentation (Fig 6F and 6G). These findings indicate that centrosomal localization of TACC and its stimulation of astral and spindle MT stability or growth is at least partially dependent on *Mms19* and CAK activity.

## Mms19 binds to MTs and stimulates MT assembly

To explore the possibility that Mms19 also functions through different activities, we prepared protein extracts from flies expressing Mms19::eGFP driven by its endogenous promoter [9]. We subjected them to immunoprecipitations (IP) and analyzed co-purifying proteins by Mass spectrometry (S1 Table). Adult wild-type flies and Imp::eGFP expressing flies, respectively, served as controls to exclude any non-specific binding to beads and GFP, respectively. Amongst the proteins exclusively bound to Mms19::eGFP were the CIA proteins Mip18, Ciao1, and Ant2, which form a complex with Mms19 to mediate Fe-S cluster delivery [6,7]. The fact that we recovered these proteins efficiently, indicated that the purification was efficient. Because *Mms19* functions on microtubules and our second control, IMP::eGFP, is also involved in MT dependent processes, we additionally inspected the data for tubulin and MT binding proteins that are enriched by the Mms19::eGFP IP compared to the wild-type control without an eGFP tag (S2 Table). This comparison revealed a clear enrichment of tubulin and several Microtubule Associated Proteins (MAPs). Some of the associated proteins were not present in the wild type control, some were present in the Mms19::eGFP and IMP::eGFP fractions, and others exclusively in the Mms19::eGFP fraction. These results, therefore, suggested that Mms19 might directly or indirectly bind to MTs.

To validate the interaction of Mms19 with tubulin, we next tested whether purified Mms19::5xHis and α/β-Tubulin dimers interact *in vitro*. Mms19::5xHis was purified from *E. coli* to avoid co-purification of MAPs, and this purified protein fraction showed a single band when separated by SDS-PAGE and stained with Coomassie Blue (Fig 7A). Mms19::5XHis was incubated at an equimolar ratio with purified porcine brain α/β-Tubulin and then bound to the Ni-NTA resin. All incubations and washing steps were carried out at 4˚C. Copurifying proteins were then assessed by western blotting (see Table 3 for details regarding the antibodies used). A band corresponding to α-Tubulin was observed when tubulin was incubated with Mms19::5xHis (Fig 7B and 7C), but tubulin alone did not bind to the resin, pointing to a direct interaction between tubulin and Mms19::5xHis.

To better understand the mechanism of modulation of the MT dynamics by Mms19, we complemented the biochemical experiments with a high-resolution optical method. Polymerized MTs were incubated at room temperature (RT) with either BRB80/solvent buffer or with Mms19::5xHis, purified upon expression in *E. coli* (Fig 7A). These samples were then visualized by negative stain electron microscopy (EM). MT fibers in general appeared to be less

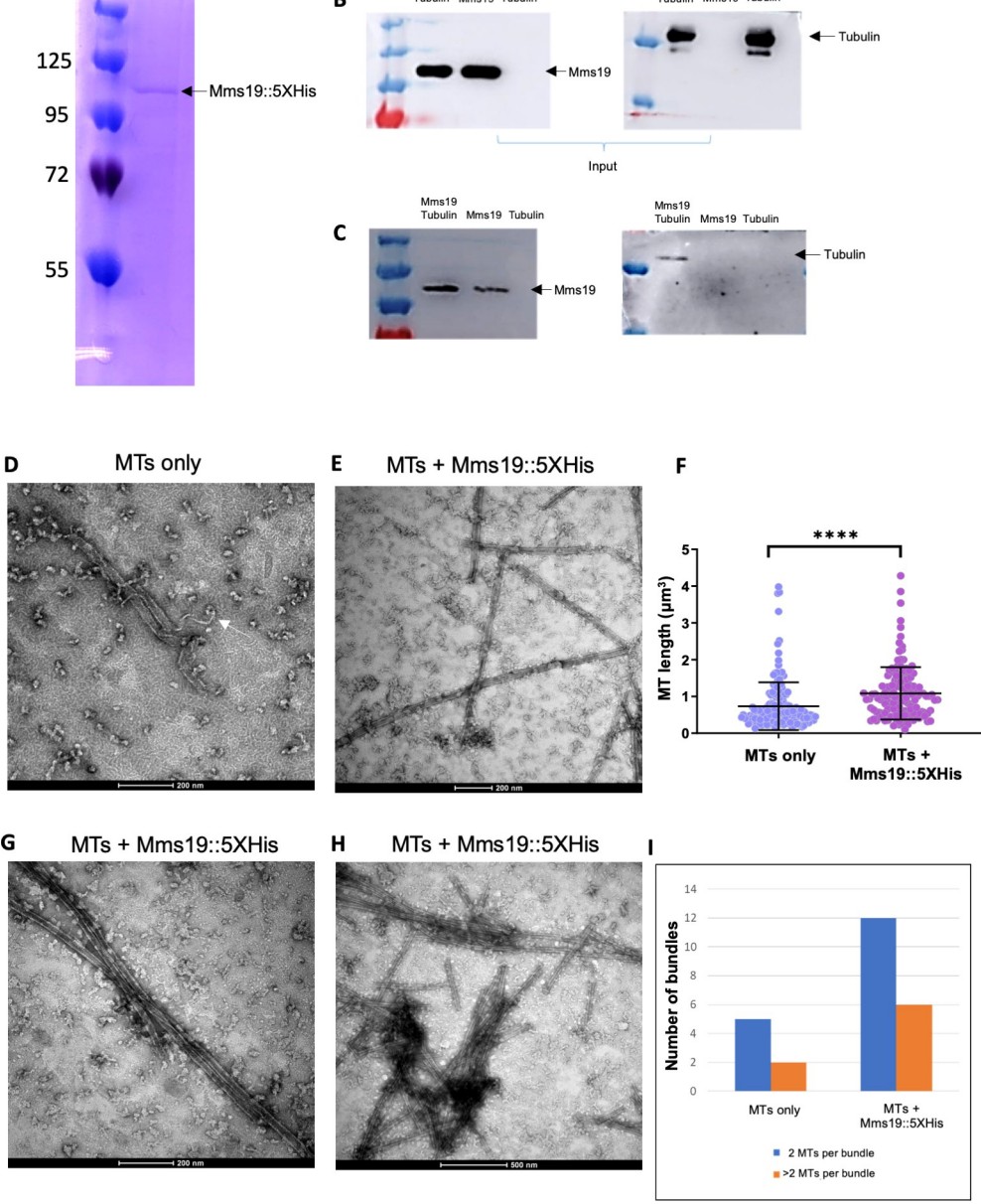

**Fig 7. Mms19::5xHis binds to tubulin and stimulates MT assembly *in vitro*.** (A) Mms19 tagged with 5X Histidine was purified from *E. coli*. A single band at approximately 100KDa is seen when the purified protein fraction is separated by SDS-PAGE and stained with Coomassie Blue. Mms19::5xHis-tubulin interaction was investigated using a pull-down assay with Ni-NTA resin. Tubulin binding to Mms19::5xHis is expected to result in co-purification of tubulin with Mms19::5xHis. (B) Input panels show tubulin added, either alone or with Mms19. (C) Tubulin was found binding to the resin only upon prior incubation with Mms19::5xHis (tubulin added alone did not bind to the resin). (D) MTs only (without Mms19::5xHis added) were visualized by negative stain EM. Here, an MT fiber can be seen depolymerizing (indicated by arrow). (E) In an MT-Mms19::5xHis mixture, discreet particles can be seen binding laterally along the surface of MTs. (F) Measuring the length of MT fibers revealed that MTs were generally longer when incubated together with Mms19::5xHis compared to MTs to which only the BRB80 buffer was added. n = 150 MTs per condition, 3 experiments, ss was calculated using student's t-test, \*\*\*\*$P < 0.0001$. MT bundles containing 2 (G) or more than 2 MTs (H) were more frequently observed upon the addition of Mms19::5xHis to MTs. The number of bundles observed in each condition is quantified in (I).

**Table 3. Antibodies used for probing western blots.**

| Primary antibodies | | | |
|---|---|---|---|
| Antibody | Manufacturer | Catalog number | Dilution |
| Rabbit anti-Mms19 | Synthesized by Genscript Inc. | Custom antibody | 1:2,000 |
| Rabbit anti-alpha-Tubulin | Abcam | Ab18251 | 1:2,000 |
| **Secondary antibodies** | | | |
| Goat anti-rabbit HRP | Thermo Fischer Scientific | 65–6120 | 1:10,000 |

stable upon the addition of BRB80/solvent alone (at RT) as some MT fibers were observed undergoing de-polymerization (Fig 7D) under these conditions. Remarkably, MTs incubated with Mms19::5xHis were comparatively longer (Fig 7E and 7F) and more bundled (Fig 7G, 7H and 7I). Furthermore, the Mms19::5xHis containing samples contained particles not found in the samples without Mms19 and these particles appeared along the lateral MT surface (Fig 7E and 7H). It thus appears that Mms19 might bind along the surface of MTs, facilitating their assembly or stabilizing MTs, and possibly stimulating inter-MT contacts.

## Discussion

*Mms19* was initially identified as a gene that acts in the nucleotide excision repair (NER) pathway. In this DNA repair pathway, it provides an Fe-S cluster to other NER enzymes. When evidence emerged about a possible NER independent mitotic activity of *Mms19* [8,9], this warranted further investigations into the precise role of *Mms19* during mitotic spindle assembly in diploid cells. We now report novel, important roles of *Mms19* for astral MT assembly, spindle establishment, and orientation, and mitotic timing in *Drosophila* NBs. We provide strong evidence that Mms19 regulates spindle assembly by promoting the activity of the Cdk activating kinase CAK, and we identified Aurora A and TACC as its downstream targets in this pathway. Further, *in vitro* data points to a CAK-independent activity where Mms19 directly binds to tubulin and thereby regulates MT assembly and stability.

The trimeric CAK complex is an essential activator of entry into and progression through the initial phase of mitosis because an activating T-loop phosphorylation by Cdk7 is required to fully activate Cdk1 [29,30]. In this activation, Mms19 seems to perform a crucial task of sequestering Xpd in order to allow CAK to fully activate Cdk1 (S6A Fig). This is evident from the rescue of imaginal disc morphology, the rescue of the TACC localization to the centrosomes, and possibly by a slight rescue of the NB spindle phenotypes by CAK overexpression in the *Mms19$^P$* background [9] (Fig 3G and 3H; Fig 6). Cdk1 along with other mitotic kinases such as Aurora A and Polo regulate a multitude of mitotic processes including centrosome maturation, bipolar spindle assembly, and mitotic checkpoint activation [31,32]. Centrosome maturation and entry into mitosis were known to be driven by positive feedback loops between Cdk1, Aurora A, and Polo [32,33]. Additionally, it was shown that Cdk1 acts upstream of Aurora A as it activates Aurora A during the G2/M transition [28]. Cdk1 was also shown previously to regulate the cytoplasmic localization of Bora, an activator of Aurora A, in *Drosophila* sensory organ precursors [34]. Our demonstration that *Mms19* is required for the localization of Aurora A to the centrosomes (S4E–S4G Fig) nicely fits into this model because *Mms19* promotes the mitotic kinases cascade through CAK-Cdk1 [9], and one function of Cdk1 is to activate Aurora A. Further, Aurora A directly phosphorylates TACC thereby triggering its centrosomal localization [26]. Given that the failure of TACC localization in the absence of *Mms19* can be rescued by CAK over-expression, we now propose TACC as a downstream target of the Mms19-CAK-Cdk1 axis. In this process, the modulation of the CAK activity seems

to be the main mode of action of Mms19 towards TACC. The best-characterized function of TACC is its interaction with Msps which leads to the stabilization of astral MTs [35]. This activity involves their recruitment to the centrosomes, and we found that not only the centro-somal localization of TACC (Fig 6) but also the one of Msps depends on *Mms19* and elevated CAK activity (S4A–S4D Fig). Additionally, it has been reported that Aurora A also mediates the timely degradation of Cyclin B and that NBs lacking functional *Aurora A* spend more time in mitosis due to delayed Cyclin B degradation [36]. Apart from the spindle assembly defects this effect could also contribute to the mitotic delay observed in *Mms19*$^P$ NBs. The identified pathway downstream of Mms19 might thus promote spindle assembly and regulate mitotic duration.

The phenomenon of centrosome asymmetry is well documented in *Drosophila* NBs where, after centriole duplication, the daughter centriole retains MT nucleation activity and is anchored to the apical cortex whereas the mother centriole generates an aster just before NEBD [37]. Interestingly, despite this asymmetry, aster formation from both centrosomes happens synchronously in WT NBs. Conversely, a striking feature of some of the *Mms19*$^P$ NBs is the delay in MT assembly from the 'mother' centrosome destined for the GMC (S5B and S5C Fig; S3 mov; S8 mov). In the example presented in S8 mov, the 'mother' centrosome seems to be unable to nucleate MTs until 7–8 minutes post NEBD. Importantly, even in this case, the spindle is bi-astral, indicating that the centrioles have duplicated normally, producing a functional pair of centrosomes. From this, we can narrow down the function of *Mms19* to the assembly of centrosomal MTs. The astral MTs are likely a main target, and their drastic loss in the *Mms19* mutants the main cause of the observed defects in centrosome migration and spindle orientation, because both these processes require contact of long astral MTs with the cortex [25,38].

It has been established that mitotic spindle assembly in mammalian cells or in *Drosophila* NBs is driven by both centrosomal MTs and MTs emanating from the chromatin [39,40]. We focused our studies mainly on the MTs emanating from the centrosomes because we were able to link the *Mms19* activity to the localization of TACC and Msps to this place. However, the formation of MTs around the chromatin might also be affected in the *Mms19*$^P$ NBs. Spindle assembly normally also initiates around the chromatin, and an MT meshwork is seen forming in the region around the chromatin [40]. In the MT re-growth assay in the *Mms19*$^P$ NBs (Fig 5B), we observed a MT mesh forming in the region around the chromatin at t = 30s. But com-pared to wild-type NBs, the length and density of MTs in this region remained abnormally short at t = 90s. Additionally, measuring the inner spindle density (Fig 3H) revealed a signifi-cant decrease in *Mms19*$^P$ NBs. As chromatin-nucleated MTs would also contribute to the MT density in the inner spindle, these results indicate defects in the assembly of MTs around chro-matin. Furthermore, live imaging of *Mms19*$^P$ NBs (S2 mov, S8 mov) showed that the overall spindle MT mesh was initially very sparse, and only after 7-8min post NEBD, a fully assembled spindle with MT density comparable with wild-type NB spindles became visible. These find-ings thus indicate that *Mms19* is needed for MT formation from both centrosomes and from around the chromatin.

CAK overexpression rescued TACC localization, but it could not fully rescue the spindle assembly/short spindle and the microcephaly phenotypes (Figs 1, 3 and 5). Earlier reports [9,41] suggested a direct interaction between Mms19 and MTs as Mms19 was shown to par-tially co-localize with spindles in mammalian cells and *Drosophila* embryos. We made similar observations in NBs expressing Mms19::eGFP (S5D Fig). Mms19::eGFP partially co-localizes with the spindles and seems to be enriched on astral MTs. Additionally, Mms19::eGFP also co-localizes with the MT bundles in the neurite of cultured neurons (S5E and S5F Fig). In this study, we report for the first time a direct interaction between Mms19 and tubulin, and we

linked this interaction to an activity of Mms19 in stimulating MT assembly and stability (Fig 7). Even though more work is needed to understand this activity, the first results on this additional function are intriguing and call for its further exploration.

Recently, it was shown that Mip18/Galla-2 and Xpd, which are binding partners of Mms19 [8,9], form a complex called 'CGX' (Crumbs/Galla-2/Xpd), and this complex recruits Kpl61F, the fly homolog of Kinesin-5, to the embryonic spindle [42,43]. Kinesin-5 activity is crucial for centrosome separation and bipolar spindle assembly [44]. Even though a direct interaction between Mms19 and Kinesin-5/Kpl61F has not been reported [43] (this work) and we do not see the typical monopolar spindle defect associated with *Kpl61F* [40] in the *Mms19*[P] NBs, it is possible that the absence of *Mms19* may compromise the activity of the CGX complex, too, and that the misregulated Kinesin-5/Kpl61F activity might contribute to the observed spindle defects.

Based on our findings and published data [9,45,46], we propose a model that outlines the CAK dependent and independent activities of Mms19 required for efficient mitotic spindle assembly in diploid cells. In the CAK dependent pathway model (S6A and S6B Fig), Mms19 competes with the CAK complex for binding to Xpd [8,9]. We hypothesize that CAK is mostly associated with Xpd during interphase, and integrated into the TFIIH complex, where its mitotic activity is inhibited. Although basal levels of CAK activity might persist, they would not be sufficient for full activation of Cdk1 (S6A Fig). During mitosis, binding of Mms19 to Xpd prevents the interaction between Xpd and the CAK complex and between Xpd and the core TFIIH. This releases the inhibition of the Cdk-activating kinase activity of CAK by Xpd and allows CAK to fully activate Cdk1 to drive mitosis [9,45]. Upon knockdown or inactivation of *Mms19*, more CAK would remain bound to Xpd and TFIIH, preventing it from optimally activating Cdk1 (S6B Fig). Nevertheless, basal levels of free CAK would remain present. Such reduced levels of CAK activity might still allow the cells to enter mitosis but would be unable to activate adequate levels of Cdk1. The low levels of *Cdk1* activity would thereby affect the downstream pathways, such as the localization of TACC and Msps through Aurora A (S6B Fig). Additionally, we now found evidence for a CAK-independent activity where Mms19 directly binds to MTs, promotes MT stability, and stimulates inter-MT contacts (Fig 7, S6C Fig).

*Mms19*, a gene initially described as a NER regulator, evidently has clear and essential additional roles as a mitotic gene and as an MT regulator. Given the critical roles it fulfills, the proper control of Mms19 expression and localization must be crucial for the proper functioning of *Mms19*. Future studies should therefore address the transcriptional and post-translational control of Mms19 expression and localization and how this impacts its functions in cell physiology, development, and diseases.

## Materials and methods

### Dissection and immunostaining of larval brains

Larval brains were dissected and stained as described [47]. Briefly, wandering third instar larvae were dissected in PBS and fixed for 15min in 4% paraformaldehyde supplemented with 0.3% Triton X-100, 100mM PIPES (pH 6.9), 1mM $MgSO_4$[2] and 1mM EGTA (to stabilize MTs). After 3 washes in PBS+0.1% TritonX-100 (PBST), primary antibodies were added (Table 2). After overnight incubation, the brains were washed 3x with PBST, stained with secondary antibodies (Table 2), and mounted on glass slides in Aqua Polymount mounting medium (Polysciences; for details on reagents and kits, see Table 4).

For MT regrowth assays, brains were dissected in Schneider's medium (supplemented with 10% fetal bovine serum) at 25˚C and incubated on ice for 30 min to depolymerize MTs. Brains

**Table 4. Kits and reagents used for these studies.**

| Reagent/Kit | Manufacturer | Catalog number |
|---|---|---|
| Aqua Poly/Mount mounting medium | Polysciences Inc | 18606–20 |
| Click-it EdU incorporation kit, Alexa Flour 647 | Thermo Fischer Scientific | C10340 |
| Paclitaxel (Taxol) | Sigma | T7402 |
| Purified porcine tubulin | Cytoskeleton Inc | T240-B |
| Schneider's Drosophila medium | ThermoFischer Scientific | 21720–024 |
| Hoechst 33342 | ThermoFischer Scientific | H3570 |
| Protein G Mag Sepharose Xtra | GE life sciences | 28967066 |
| Ni-NTA agarose | Qiagen | 30210 |
| Collagenase I | Sigma-Aldrich | C0130-100MG |
| Phenylmethylsulfonyl fluoride (PMSF) | Sigma-Aldrich | 10837091001 |
| cOmplete, Mini, EDTA-free Protease Inhibitor Cocktail | Sigma-Aldrich | 4693159001 |
| Concanavalin A | Sigma-Aldrich | C7898 |
| Formamide | Sigma-Aldrich | F9037-100ml |
| Uranyl acetate | Electron microscopy sciences | 22400 |
| **1,4-Piperazinediethanesulfonic acid, Piperazine-1,4-bis(2-ethanesulfonic acid), Piperazine-N,N′-bis (2-ethanesulfonic acid) (PIPES)** | Sigma-Aldrich | P6757 |
| **5' Cy5 labeled Oligonucleotide probe:** 5' Cy5-AACACAACACAACACAACACAACACAACACAACAC | Microsynth AG | ChrII |
| Aphidicolin | Sigma-ALdrich | A0781 |

were then incubated for different time points in a 25˚C water bath, followed by fixation and immunostaining. This experiment was performed three times and 20 spindles were analyzed in each iteration. Slides were imaged on a Leica TCS-SP8 microscope (Leica Microsystems) equipped with a 63X, NA 1.4 Plan Apochromat objective. Images were acquired using LAS X software and analyzed using Fiji/ImageJ [48] (see Table 5 for details regarding the special software used).

## Measuring brain compartment volumes

Whole brain lobes were immunostained with antibodies against Miranda (Mira) and phospho-Histone 3 (pH3), and DNA was visualized with Hoechst 33342. Z stacks were acquired with a TCS-SP8 confocal microscope on a 63X, 1.4NA Plan-Apochromat objective with 1μm spacing between each optical section. Segmentation, volume measurement, and 3D reconstruction was performed by using the TrackEM2 plugin in Fiji [49].

## MARCM crosses

For generating mosaic clones we used the methods described previously [50] with modifications. Briefly, the driver stock *hs-flp; tub-Gal4, UAS-mCD8::GFP/CyO, actin::GFP; FRT82B, tub-Gal80/TM6*, Tb was crossed to *+; FRT82B, Mms19^P^/TM6, Tb*. GFP-Balancer negative 24hr

**Table 5. Software used.**

| Software | Source | Version |
|---|---|---|
| Fiji (ImageJ) | https://imagej.net/Fiji | - |
| Leica Application Suite (LAS X) | Leica microsystems | - |
| PRISM | Graph pad software | Version 5 |

old larvae were selected and heat shocked at 37˚C in a glass vial submerged in a water bath for 15min. Larvae were then returned to 25˚C and the brains of the non-Tubby larvae were dissected 48hrs later.

## Live imaging

Brains expressing EB1::GFP were dissected and mounted on stainless steel chambers as described in [51]. Brains were then imaged using a 100X, NA 1.3 oil immersion objective on a Visiscope Spinning disk microscope (Visitron GmbH) fitted with Nikon Ni2 stand, a CSU-W1 scanner unit, and a Photometrics Evolve 512 EMCCD camera. Images were acquired for 60 seconds with 500ms time intervals at 200ms exposure at 60% laser power (488nm). To track the particle velocities, the particles were manually traced in Fiji/ImageJ [48]. Probably due to limited resolution, it was not possible to unambiguously account for merging or splitting events, and therefore only particles with a linear trajectory, which did neither split nor merge, were analyzed. At least 4–5 particles from each spindle were analyzed from a total of 11 cells per genotype. For surface glia quantification, at least 7–8 particles were analyzed from each of the 15 brains. Stacks were exported to.AVI movies at 14 frames per second.

To measure the NEBD to Anaphase B duration, EB1::GFP expressing brains were mounted as described above and imaged at 40% laser power with a 63X, 1.3 NA objective of a Nikon W1 LIPSI spinning disk microscope fitted with a Photometrics Prime 95B CMOS camera. Movies were acquired on Nikon's NIS elements software for 2hrs with an interval of 1 min at 200 ms exposure. Z-stacks were acquired simultaneously with 2μm distance between successive optical sections. Movies were analyzed and processed with Fiji/ImageJ. Stacks were exported to.AVI movies at 12 frames per second.

## Quantification of spindle orientation

The orientation of the mitotic spindle was examined with respect to the basal Mira crescent. A reference line was drawn passing approximately through the center of the Mira crescent and the angle between the spindle and this reference line was determined using the ImageJ angle tool.

## EdU incorporation

Click-it EdU kit (Invitrogen) was used to measure EdU incorporation. Brains were dissected in PBS and incubated in Schneider's medium supplemented with 10μM 5-Ethynyl-2'-deoxyuridine (EdU) for 2hrs at 25˚C. EdU is an analog of Thymidine and is incorporated by S phase cells during DNA replication. EdU is then detected due to its binding to a dye-azide conjugate. Brains were subsequently fixed with 4% PFA and incubated with primary antibodies (anti-Mira to mark NBs and anti-pH3 to mark mitotic cells) and secondary antibodies. Subsequently, they were processed for EdU detection following the manufacturer instructions.

## Preparation of whole-fly extract

1g of flies were collected in Eppendorf tubes and frozen in liquid nitrogen. Frozen flies were crushed into a fine powder using a pre-cooled mortar pestle. The powder was incubated in lysis buffer (25mM Hepes, 150mM NaCl, 1mM EDTA, 0.1% TritonX-100, 1mM phenyl-methylsulfonyl fluoride (PMSF), 1 complete EDTA-free protease inhibitor tablet (Roche/Sigma-Aldrich) for 30 min and then centrifuged in an Eppendorf tube for 30 min at 16,000 g and 4˚C. The supernatant was saved and snap-frozen in liquid nitrogen.

## Immunoprecipitation to prepare extract for Mass Spectrometry

ProteinG-Mag Sepharose (GE) beads were washed 3X with PBS, incubated for 2hrs with anti-GFP antibody (3E6, provided by Anne Marcil). Beads were then incubated with the crude extracts for 6hrs at 4°C. Following 3 washes with the wash buffer (25mM Hepes, 150mM Nacl, 1mM EDTA, 1mM PMSF, 1 tablet complete EDTA free protease inhibitor tablet), bound proteins were eluted by 15 min incubation in urea elution buffer (6–8 M Urea, 20 mM Tris pH 7.5, and 100 mM NaCl) or glycine elution buffer (100mM Glycine, pH 2.6. These eluates were neutralized by adding 150mM Tris-Cl, pH 8.8).

## Mass-spectrometry

Eluted proteins in 8M urea were processed essentially as described by Engel and colleagues [52]. Briefly, proteins were reduced by the addition of 1/10 volume of 0.1 M DTT and incubated for 30 min at 37°C, followed by alkylation with a five-fold molar excess of iodoacetamide and incubation for 30 min at 37°C. Proteins were precipitated at -20°C by the addition of 5 volumes cold acetone and incubation for 30 min at -20°C. All liquid was carefully removed, and the pellet dried in ambient air for 15 min before reconstitution of the proteins in 8 M urea, 50 mM Tris-HCl pH 8.0 to a final protein concentration between 0.2–0.3 mg/mL. Protein concentration was determined by Bradford assay. An aliquot corresponding to 5 μg protein was diluted to a final urea concentration of 2 M urea with 20 mM Tris-HCl pH 8.0, and 2 mM CaCl2. Proteins were digested by trypsin (1:50 (w/w) trypsin/protein ratio) for 6 hours at 37°C. The digests were acidified with TFA (1%) and analyzed by LC-MS/MS (EASY-nLC 1000 coupled to a QExactive HF mass spectrometer, ThermoFisher Scientific) with three repetitions injecting an aliquot of 500 ng protein. Peptides were trapped on an Acclaim PepMap100 C18 pre-column (3μm, 100 Å, 75μm x 2 cm, ThermoFisher Scientific, Reinach, Switzerland) and separated by backflush on a C18 column (3μm, 100 Å, 75μm x 15 cm, Nikkyo Technos, Tokyo, Japan) by applying a 40 min gradient of 5% acetonitrile to 40% in water, 0.1% formic acid, at a flow rate of 300 nl/min. Peptides of m/z 400–1400 were detected at a resolution of 60,000 m/z 250 with automatic gain control (AGC) target of 1E06 and maximum ion injection time of 50 ms. A top fifteen data-dependent method for precursor ion fragmentation was applied with the following settings: resolution 15,000, AGC of 1E05, maximum ion time of 110 ms, charge inclusion of 2+ to 7+ ions, peptide match on, and dynamic exclusion for 20 sec, respectively.

Fragment spectra data were converted to mgf with ProteomeDiscoverer 2.0 and peptide identification made with EasyProt software searching against the forward and reversed UniprotKB *Drosophila melanogaster* protein database (Release 2016_11), complemented with commonly found protein sequences of contaminating proteins, with the following parameters: parent mass error tolerance of 10 p.p.m., trypsin cleavage mode with three missed cleavages, static carbamidomethylation on Cys, variable oxidation on Met and acetylation on protein N-terminus. On the basis of reversed database peptide spectrum matches, a 1% false discovery rate was set for acceptance of target database matches, and only proteins with at least two different peptide sequences identified were allowed.

## Immunoprecipitation/pull-down assays

50μg of purified Mms19 (Tagged with 5X Histidine at C-terminus, synthesized by Genscript Inc and solubilized in 20mM Tris, 150mM NaCl, 0.5 M Arginine) was incubated with 50μg purified porcine tubulin (Purchased from Cytoskeleton Inc) at 4°C for 2 hrs. This mixture was subsequently incubated with Ni-NTA agarose (equilibrated with 20mM Tris-Cl and 250mM NaCl) for 1 hr at 4°C. The beads were then washed three times with wash buffer containing 20mM Tris, 250mM NaCl, and 20mM Imidazole, and the bound proteins were eluted by

adding elution buffer (20mM Tris, 250mM NaCl, and 500mM Imidazole) to the resin on ice for 10 min. The eluate was then analyzed by probing Western blots with anti-Mms19 antibodies and rabbit anti-alpha-Tubulin antibodies.

## Neuronal *in vitro* cultures

Dissociated brain cells were cultured *in vitro* according to a protocol previously described [53]. Briefly, 15–20 third instar larvae were dissected in PBS and washed 3 times with Rinaldini solution (800 mg NaCl, 20 mg KCl, 5 mg $NaH_2PO_4$, 100 mg $NaHCO_3$, 100 mg glucose, in 100 ml distilled water). The brains were then incubated in 0.5% collagenase I in Rinaldini solution for 60 min and subsequently washed 4 times in Schneider's medium. The treated tissues were then dissociated by pipetting 100–200 times. The resulting cell mixture was passed through a 40μm mesh to remove cell clusters, and the brain cells were then incubated for 24hrs at 25°C. After 24hrs incubation, the cells were plated on concanavalin A-coated coverslips, fixed and stained.

## Preparation of MTs for EM

To obtain polymerized MTs, 20μM tubulin was incubated in the presence of 10μM taxol at 37°C for 30min. The sample was then subjected to ultra-centrifugation for 10min at 100,000 g at 4°C in a Beckman Airfuge ultracentrifuge. The supernatant containing un-polymerized dimers was then removed and the pellet was reconstituted with either 15μl BRB80/solvent (BRB80 components: 80mM PIPES buffer, 1mM EGTA, 2mM $MgCl_2$ + Mms19 solvent constituents: 20mM Tris-CL, 150mM NaCl, 0.5M Arginine) or 15 μl of 0.5μM Mms19::5xHis in BRB80/solvent.

## Negative stain EM

5 μl of samples were applied on glow discharged, carbon-coated copper EM grids for 1min. The excess sample was then washed off by dipping the grid in milli-Q water. The sample on the grid was then fixed/negatively stained with 2% Uranyl acetate for 30 sec and then the excess fluid was removed by using filter paper. The samples were imaged at a nominal magnification of 63,000x or 87,000x on a FEI Tecnai Spirit EM operated at 80eV and fitted with a digital camera. MT length and number of MT bundles were quantified from images obtained from three independent experiments.

## Fluorescent *in situ* hybridization (FISH)

The protocol for FISH was adapted from [54] with minor modifications. Briefly, third instar larval brains were dissected in PBS and fixed with 4% PFA. The brains were then washed 3 times for 10min each with 2xSSCT (0.3M Sodium Chloride, 0.03M Sodium Citrate and 0.1% Tween 20) followed by 10min washes respectively with 2xSSCT/20% Formamide, 2xSSCT/40% Formamide, 2xSSCT/50% Formamide. 100ng of the oligonucleotide probe: 5' Cy5-AA-CACAACACAACACAACACAACACAACACAACAC that binds to a specific region on the 2<sup>nd</sup> chromosome [54] was then added to the Hybridization buffer (20% dextran sulfate, 2xSSCT, 50% Formamide) and this solution was incubated with the brains in a PCR tube. The probes were then denatured at 92°C for 3 min and then allowed to anneal with the chromosomal DNA overnight at 37°C. The sample was then washed thrice at 37°C with the following solutions for 20min each: 2xSSCT/50% Formamide, 2xSSCT/40% Formamide, 2xSSCT/20% Formamide. After two more washes with 2xSSCT, the sample was stained with Hoechst,

mounted using Aqua-poly mount, and imaged on a Leica SP8 confocal microscope with a 63x objective.

### Statistical analysis

Data were analyzed using Graph Pad prism 5.0 software. Means of two groups were compared and significance calculated by using unpaired student's t-test. Percentages were compared and significance was calculated by using Fisher's exact test. Multiple groups were compared, and significance calculated using the Kruskal-Wallis test and the Dunn's post test.

### Supporting information

**S1 Fig. *Mms19^P^* NBs do not display elevated levels of aneuploidy.** Wild type, *Mms19^P^*, and da>CAK, *Mms19^P^* brains were fixed and fluorescent *in situ* hybridization was performed on them. Cy5-labeled DNA probes that specifically bind to regions on the 2^nd^ chromosome were used to determine the number of 2^nd^ chromosomes. (A) The signal is seen as 2 dots in the WT NBs corresponding to the diploid state of the cell. (B) An example of an aneuploid *Mms19^P^* NB showing 3 dots. (C) Percentages of diploid (green) aneuploid (red) cells in each genotype were quantified. Aneuploid cells in *Mms19^P^* (and da>CAK, *Mms19^P^*) NBs were extremely rare and not significantly elevated ($P = 0.1493$). WT, *Mms19^P^*: n = 200, da>CAK, *Mms19^P^*: n = 100. SS was calculated using Fisher's exact test.
(PDF)

**S2 Fig. Higher fraction of *Mms19^P^* NBs in mitosis.** (A) NBs were classified into 1) G1/G0 phase if they did not stain for either EdU or pH3; 2) S phase if the NBs stained positively for EdU; 3) Mitotic phase for NBs staining positively for pH3 (independent of whether they stained for EdU or not). NBs in each phase were counted per brain lobe and this data was represented as percentage of total NBs in this lobe (e.g. if in one brain lobe 20 out of 100 NBs were EdU positive, then 20% cells were classified as in S phase). The percentages for each phase were compiled and compared per brain lobe across the 4 genotypes. Scatter dot plot charts represent the percentage of cells in (B) G1/G0 phase, (C) S phase and (D) mitotic phase. n = 30 brain lobes per genotype, experiments. SS was calculated using Kruskal-Wallis test, columns compared using Dunn's post test, ****($P<0.0001$), ***($P<0.001$), *($P<0.05$). Scale = 5μm
(PDF)

**S3 Fig. *Mms19* is cell autonomously required to maintain normal cell numbers in MARCM clones.** (A)-(C) In order to study cell cycle progression in a single NB lineage, we used the Mosaic Analysis with a Repressible Cell Marker (MARCM) technique [50]. This technique utilizes the UAS-GAL4-GAL80 system and the FLP-FRT recombination system. With this technique, a population of cells arising from the same progenitor can be specifically labeled. Additionally, the progenitor cell can carry a mutation along with a GFP marker. Defects in this cell, along with its progeny can be analyzed in an otherwise wild-type background. (B) MARCM clones were induced in NBs in 24hrs old larvae. These larvae were dissected after another 48hrs to determine the number of cells per clone in *Mms19^P^* and wild-type control clones. (C) The graph shows a significant reduction in the numbers of cells in mutant clones. SS was determined by an unpaired t-test (**$P<0.01$), scale = 5μm, n = 60 clones from each genotype.
(PDF)

**S4 Fig. *Mms19* is necessary for centrosomal localization of Aurora A and Msps in NBs.** (A, C) WT and da>CAK, *Mms19^P^* NBs showing Msps localization on centrosomes and spindles. (B) In *Mms19^P^* NBs, Msps does not concentrate on centrosomes. (D) To quantify the

centrosomal accumulation of Msps, an analysis similar to that done in Fig 6 was performed. SS was calculated using Kruskal-Wallis test, columns were compared using Dunn's post test, ***($P<$0.001), *($P<$0.05), scale = 5μm. WT, n = 25 NBs; *Mms19*[P], n = 30 NBs; da>CAK, *Mms19*[P], n = 9 NBs, 2 experiments. (E) Aurora A signal is enriched in the spindle pole region in metaphasic WT NBs (indicated by arrows) but seems to be depleted from the spindle pole region of *Mms19*[P] NBs (F). (G) The scatter plot represents the ratio of the fluorescent intensity of Aurora A on the centrosome to the background fluorescent signal on the spindles. N = 28 cells per genotype, 2 experiments. Columns were compared using unpaired Students' t-test, ****($P<$0.0001).
(PDF)

**S5 Fig. MT assembly defects in *Mms19*[P] NBs and Mms19::eGFP localization in NBs and in neurons.** (A) WT NBs assemble a bipolar spindle 2–3 minutes after NEBD. On the other hand in some *Mms19*[P] NBs, (B, C) we observed a delay in MT assembly from one centrosome (indicated with arrows) and bipolar spindle assembly in these cells took on average 7–8 mins after NEBD. The centrosome which showed a delay in MT assembly was always inherited by the GMC. (D) Mms19 localization in NBs was determined by staining Mms19::eGFP, *Mms19*[P] NBs with anti-GFP antibodies. Although the Mms19::eGFP signal appears ubiquitous in the cytoplasm, we observe an enrichment on astral MTs (indicated by arrows). Scale = 5μm, n = 30 NBs, 2 experiments. (E) Neurons expressing Mms19:eGFP in the *Mms19*[P] background were stained with anti-GFP antibody to determine the localization of Mms19 in neurons. Mms19:eGFP signal co-localizes with α-Tubulin in the neurite. Scale = 5 μm, n = 30 neurons, 2 experiments. (F) No signal was observed in WT neurons stained with anti-GFP antibody, thus ruling out any non-specific signal by the anti-GFP antibody.
(PDF)

**S6 Fig. Model for the function of Mms19 towards MTs.** (A) During interphase, much of CAK is bound to the core TFIIH via Xpd. Even though basal levels of free CAK (shown above the TFIIH in faint colors) exist, this activity is below the required threshold to push cells into mitosis. During mitosis, Mms19 binds to Xpd, and thereby releases CAK and ensuring that sufficient CAK activity can drive mitosis via activation of Cdk1 and its downstream targets including Aurora A, TACC, and Msps. (B) Downregulation of Mms19 by mutations or knock-down allows Xpd to associate with CAK and core TFIIH, thereby targeting Cdk7 activity away from the mitotic targets and towards transcriptional targets like the PolII-CTD [10]. Though basal levels of CAK activity remain in this case, they are not able to bring about optimal activation of Cdk1, and therefore, when cells enter mitosis, this results in spindle assembly defects and mitotic delays. (C) Mms19 binds to MTs and appears to promote MT assembly, stability, and bundling. This novel activity of Mms19 could potentially contribute to establishing the extended MT structures in the mitotic spindle.
(PDF)

**S1 Table. The table lists proteins found to exclusively co-purify with Mms19::eGFP.** CIA proteins, which are already known to form a complex with Mms19, are highlighted in Red. Microtubule associated proteins are highlighted in Green.
(PDF)

**S2 Table. Proteins that bound to the anti-GFP antibody coated beads from all three fly extracts are listed.** These include different isoforms of α-and βtubulin that were identified in all extracts. However, the PMSS scores are higher for the tubulin isoforms immuno-precipitating from the Mms19::eGFP extract. This indicates that tubulin was enriched in the Mms19::

eGFP fraction compared to the wild-type control.
(PDF)

**S3 Table. List of data points used to build the graphs and the tests to determine statistical significance for each data set and the corresponding P values.**
(XLSX)

**S1 Movie. Analysis of mitosis duration in WT NB: WT brains expressing EB1::GFP were dissected, mounted on a stainless steel chamber (Cabernard et al, 2013) and NB mitosis was imaged with a 63x objective on a spinning disk confocal microscope.** Mitosis duration was measured from NEBD onset, that starts from 0 min until cytokinesis. Scale = 5μm
(AVI)

**S2 Movie. Analysis of mitosis duration in *Mms19^P* NB: Mitosis was visualized in *Mms19^P* brain NBs expressing EB1::GFP.** On average, *Mms19^P* brain NBs took twice as long as WT NBs to finish mitosis. In this presented case, the NB completes mitosis in 22min when measured from NEBD onset until cytokinesis. Scale = 5μm
(AVI)

**S3 Movie. Spindle assembly defect in *Mms19^P* NB: In around 10% of EB1::GFP expressing *Mms19^P* brain NBs, the spindle starts assembling before the centrosomes have migrated to the opposite sides, forming a 'kinked' spindle at 4min post NEBD onset.** Scale = 5μm.
(AVI)

**S4 Movie. Analysis of EB1::GFP labelled MT velocity in WT NBs: In order to determine the speed of growing MT tips, WT brains expressing EB1::GFP were dissected, mounted on a stainless steel chamber (Cabernard et al, 2013) and spindles were imaged with a 100x objective on a spinning disk confocal microscope.** Images were acquired at an interval of 500ms for 1min. Scale = 5μm.
(AVI)

**S5 Movie. Analysis of EB1::GFP labelled MT velocity in *Mms19^P* NBs: Live imaging was performed on *Mms19^P* brain NB spindles to determine the velocity of growing MT tips.** Images were acquired at an interval of 500ms for 1min. The spindle shown here appears to tilt repeatedly, perhaps due to defects in astral MT assembly (Fig 2, Fig 6). Scale = 5μm.
(AVI)

**S6 Movie. Measuring MT plus tip speeds in post mitotic surface glia of WT brains: In order to determine whether *Mms19* affects MT growth in post mitotic cells, WT brains expressing EB1::GFP were dissected, mounted on a stainless steel chamber (Cabernard et al, 2013) and MT growth in surface glia was imaged with a 100x objective on a spinning disk confocal microscope.** Images were acquired at an interval of 500ms. Scale = 5μm.
(AVI)

**S7 Movie. Measuring MT plus tip speeds in post mitotic surface glia of *Mms19^P* brains: MT growth in EB1::GFP expressing *Mms19^P* surface glia was visualized using spinning disk confocal microscopy.** Images were acquired at an interval of 500ms. Scale = 5μm.
(AVI)

**S8 Movie. Spindle assembly delay in *Mms19^P* NBs: Mitotic spindle assembly was visualized in *Mms19^P* brain NBs expressing EB1::GFP.** The spindle is bi-astral, i.e. contains duplicated centrioles but MTs are observed to emanate initially only from one centrosome. A fully

assembled bipolar spindle is observed only 7-8min after NEBD.
(AVI)

## Acknowledgments

We would like to thank Alex Bird, Carlo Largiader, and our group members for helpful discussions and feedback. Our thanks also go to Regis Giet, Bruno Bello, Claudio Sunkel, Hiro Ohkura, Anne Marcil, Jordan Raff, Jurgen Knoblich, the Bloomington Stock Center (University of Indiana), DSHB (University of Iowa) for providing antibodies and fly stocks, and to Fly-Base for excellent community support. The authors also wish to thank Manfred Heller, Sophie Braga and the Proteomics Mass Spectrometry Core Facility at the University of Bern, Yury Belyaev of the Microscopy Imaging Center (University of Bern), and Beat Haenni and Benoit Zuber for their EM services and support.

## Author Contributions

**Conceptualization:** Rohan Chippalkatti, Boris Egger, Beat Suter.

**Data curation:** Rohan Chippalkatti.

**Formal analysis:** Rohan Chippalkatti, Boris Egger, Beat Suter.

**Funding acquisition:** Beat Suter.

**Investigation:** Rohan Chippalkatti.

**Methodology:** Rohan Chippalkatti, Boris Egger, Beat Suter.

**Project administration:** Beat Suter.

**Resources:** Rohan Chippalkatti, Boris Egger, Beat Suter.

**Software:** Rohan Chippalkatti.

**Supervision:** Beat Suter.

**Validation:** Rohan Chippalkatti.

**Visualization:** Rohan Chippalkatti.

**Writing – original draft:** Rohan Chippalkatti, Beat Suter.

**Writing – review & editing:** Rohan Chippalkatti, Boris Egger, Beat Suter.

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
