## [Decision Letter · Decision Letter 0]

7 Jul 2020

Dear Dr Suter,

Thank you very much for submitting your Research Article entitled “Mms19 promotes spindle microtubule assembly in neural stem cells through two distinct pathways” to PLOS Genetics. Your manuscript was fully evaluated at the editorial level and by three independent peer reviewers. The reviewers appreciated the attention to an important problem, but raised  important concerns about the current manuscript. Based on the reviews, we will not be able to accept the manuscript, at least in its current version. However, we would be willing to review again a much-revised version including experimental work. We cannot, of course, promise publication at that time. The most important concerns are:

Reviewer 2 and 3 found that additional and more detailed descriptions of Mms19 phenotype would bring a better understanding of Mms19 functions. Reviewer 2 mentioned a more precise measurement of cell cycle length. Reviewer 3 made several important suggestions including further characterization of Mms19 phenotype at the cellular level but also at the level of the organ. It could be achieve by additional quantifications and/or additional genetic interactions.In vitro experiments characterizing Mms19 direct interactions with microtubules. Reviewer 2 made some strong criticisms on experiments shown in Figure 7. One possibility would be to tone down your conclusions and/or remove some of these experiments, or perform additional experiments to answer reviewer’s 2 comments.

If you decide to revise the manuscript for further consideration at PLOS Genetics, please aim to resubmit within the next 60 days, unless it will take extra time to address the concerns of the reviewers, in which case we would appreciate an expected resubmission date by email to plosgenetics@plos.org.

[LINK]

We are sorry that we cannot be more positive about your manuscript at this stage. Please do not hesitate to contact us if you have any concerns or questions.

Yours sincerely,

Jean-René Huynh

Associate Editor

PLOS Genetics

Gregory P. Copenhaver

Editor-in-Chief

PLOS Genetics

Reviewer's Responses to Questions

**Comments to the Authors:**

Reviewer #1: In this manuscript Chippalkatti and co-workers present a thorough analysis of the Mms19 protein and its function in regulating microtubule polymerisation using neural stem cells of the developing fly brain as a model system. Using genetics, quantitative live cell imaging, immunoprecipitation and mass spectrometry analysis as well as Tubulin binding assays the authors report that Mms19 plays a dual role in regulating the morphology of the mitotic spindle. Mms19 is known to be required for nucleotide excision repair. The new roles uncovered here, suggest that Mms19 also plays an important role in microtubule regulation. The study finds that mms19 mutants have smaller brains due to malformation of the optic lobes of the brains. The study then goes on to measure the cellular phenotypes focusing on microtubules of neuroblasts in the central brain. Using these cells, the study reports that one role of MMs19 is to stimulate microtubule nucleation from the centrosome by controlling the recruitment of TACC/Msps, providing novel regulatory insights into centrosomal regulation. In addition, the study reports that Mms19 directly binds to microtubules and is likely to regulate microtubule stability, which could explain the observed phenotypes. The rationale is well explained, and the results are well written. The experimental design appears to be sound and the interpretations supported by experimental evidence.

So, it appears that Mms19 regulates mitotic microtubules as well as postmitotic microtubules. This slows down mitosis in rapidly dividing cells causing phenotypes and is likely due to a role in reinforcing robustness and stability in the mitotic spindle apparatus. To me this makes sense. The findings are also of general interest to a readership such as that of PLOS Genetics and I would in principle support publication, but the authors may wish to look a few points that I listed below.

The manuscript needs some work on figures and their legends in terms of error bars, scale bars and would benefit from some further quantification (see below). I would also try to avoid reporting results in the discussion (e.g. line 395 onwards). The discussion could be improved, working out better the part on the interpretation of how Mms19 works in the cell biological level and another to discuss the broader relevance. Perhaps not all detail present right now is really necessary (I appreciate all the work that has gone into this study, but the ubiquitination part is perhaps not necessary here, I would think that the MS results are also not necessary either, as the results have not really been followed up, they are definitively interesting though).

Comments:

Major:

The figure legends could be improved by clearly stating the number of independent biological repeats in all cases. Some of the graphs also miss error and scale bars and some essential statistical tests and further quantifications are missing, that might be important to include.

Other:

Fig 1

mms19 brains are smaller, NB number the same. How is that? Explanation: mitosis takes longer (mitotic index EdU + live imaging data) and logically MARCM clones produce less offspring. Rescued by Mms19:eGFP, while overexpression of CAK does not. Convincing

Comment: A lot of mitosis occurs in the neuroepithelium in developing brains. Da>CAK, Mms19p (Fig 1A) is the OL more damaged. i.e. accelerated cell cycle harmful in this context to this tissue or unlucky picture? So, the brain morphology appears to be largely damaged by faulty neuroepithelial processes, perhaps mentioned that in the manuscript?

Fig 1D is only referring to NBs in the “red shaded” area I guess, if so clarify.

Line 917: Ph3 labels also dividing GMCs as well as neuroepitheial cells in the OL

Fig 2+3+4

Spindle assembly is driven by centrosomal microtubule nucleation, but also by centrosome independent microtubule nucleation pathways. For instance, cells without centrioles form spindles, yet lack astral microtubules. It is a bit of a shame that this has not been looked into in more detail. Do the authors think that Mms19 plays a role in stabilizing a microtubules or just one pool (astral versus main spindle?)

Fig 3G, what exactly is quantified here is unclear. It reads like this is used to measure spindle length versus cell diameter to quantify spindle length. Provide an example of a fixation of astral microtubules in the mutant. Fig 3 B’: the signal is hard to detect. Perhaps a more accurate way would have been to look at the EB1 data in live? Less EB1 comets from centrosomes? Do spindles first form over chromatin and the centrosomal microtubule nucleation is slowed? The timing of these two events looks like what is going wrong. This is perhaps something that could be discussed? It would also be nice to show a plot of the actual tracking data, the videos are a bit noisy. Do they see evidence for EB1 coming from the centrosomes towards the metaphase plate and comets going outwards from the metaphase plate? The point that might deserve discussion is whether Mms19 might be regulating microtubule nucleation from chromatin or from existing microtubules or help stabilize this process? Do Mms19 cells have apical microtubule asters in interphase, or is it purely mitotic as the CAK axis suggest, the authors must have these results in their data sets (live imaging)?

Overall, however, the weakened spindles are convincing and the experiments to probe into this problem appear sound. Perhaps revise some wording in the figure legend of Fig 3. (e.g. line 949).

Fig 4 C,D is a perhaps bit redundant with Fig 2, without quantification?

Fig 5:Line 287: Interestingly, whereas the Mms19::eGFP fusion protein was able to rescue this phenotype, CAK overexpression was unable to do so (Fig 5C-E).

I am not sure, but does the quantification in Fig 5E really show rescue? The statistical test is missing at least. From the micrographs I would think it looks pretty well rescued. Also, for CAK the test is missing to support that it does not rescue.

Fig 7 . Mms19 binding to Tubulin interesting, but essential quantification is missing for microtubule bundling effect in the EM data, this could perhaps be improved.

Reviewer #2: Mms19 promotes spindle microtubule assembly in neural stem cells through two distinct pathways. By Chippalkathi, Egger and Suter.

In this study, the authors have characterized in vivo, using Drosophila, the phenotype of mms19 mutant during brain development. They have shown that the mms19 mutation causes microcephaly and impairs NB proliferation caused by a mitotic delay. Interestingly, mms19 cells exhibit mitotic spindle assembly defects and failure to recruit the D-TACC/Msps complex at mitotic centrosomes. This particular defect can be rescued by overexpressing the CAK complex. The authors also describe that their mitotic phenotype, may to some extend also be caused by a direct effect of Mms19 protein on microtubule polymerisation or stabilisation, as suggested by in vitro experiments with pure tubulin and MTs.

In one hand, I think this is potentially interesting story. On the other hand, the way some of the experiments were done does not convincingly support the conclusions that are inferred.

In particular, the direct effect of Mms19 on MTs remains to be shown.

Major points

1-Lane 183. There is confusion in the mitotic duration analysis. Mitotic timing is the time between NEBD and anaphase onset (which reflects the time to assemble a spindle and satisfy the spindle checkpoint). Cytokinesis completion is not appropriate to determine the mitotic timing, and cannot be determined by using EB1-GFP. I recommend analyzing all the movies to make new figures and calculate real mitotic duration (which is between 5 and 7 min in control NBs). Ideally, a double SAC+ mms19 (mad2) would shorten the mitotic timing and validate that the delay in M phase is caused by SAC activation.

2- I am not convinced by the in vitro studies presented here.

-Indeed, many proteins show the ability to bind tubulin in vitro (especially using such high concentration of proteins) that reflect unspecific interactions, aggregations. Can we see how “pure” is recombinant (His)6-Mms19 on a coomassie gel ? Is the prep contaminated by other proteins ?

-Moreover the measured 340 nm OD for the tubulin polymerisation experiment rather suggests aggregation than polymerisation. Why tubulin doesn’t not show spontaneous polymerisation on its own ? In a classical turbidity assay with 40 microM tubulin, the OD should reach 0.4.

-Interestingly the authors suggest that MTs are decorated by discrete particules in their EM pictures (Figure 7) supporting the hypothesis that Mssp19 would be a microtubule associated protein. These EM experiments frequently leads to artifacts (depending on protein purity).

-The fact that Mms19 would be a MAP is not demonstrated in this manuscript :

-Interestingly, the authors do have a functional Mms19-GFP transgenic line. How is the protein localized in vivo ? Is it associated with MTs or spindles ? Their previous work published in Developmental Biology (2018), suggested it is a cytoplasmic protein. May be better pictures could be provided here with live NBs expressing Mms19-GFP in the mutant background.

- One would expect for a stabilizing protein/MAP that overexpression would lead to MT stabilization in vivo. Is it possible to overexpress Mms19 and analyze MT networks ?

-It is indeed tempting to speculate, given the lower MT polymerisation speed of glial cells that Mmsp19 is involved in the control of MT dynamics but these brains are heavily affected by the loss of mms19 and it could be a secondary effect. The same remark can be made for the lack of neurites extensions in mms19-cultured neurons.

To conclude, there are no convincing evidences that MMs19 regulates MTs on its own and is responsible for the second pathway regulating spindle assembly.

3-Epistatic experiments show that in mms19 the main problem is a defective CDK1 activation and spindle assembly defects (likely because CAK remains sequestrated by XPD). This is not supported by figure S1 that reveals that mms19 mutant displays lower number of G2 cells and higher mitotic cells. We expect higher numbers in both categories.

However, despite the absence of Mms19, cells manage to enter in mitosis suggesting Cdk1 can be still activated (therefore the model presented in figure S3 C is wrong, or at least too simple and should probably include other triggers of CDK1 activation).

I feel that the mitotic phenotype phenotype seen here may be caused by a weaker cdk1 activation (as suggested by the fact that CAK overexpression rescues D-TACC recruitment). This could be challenged experimentally by FRET probes for CDK1 (but these are complicated and time experiments). Alternatively, immunostaining with phosphoantibodies for known CDK1 targets could be performed.

I am also surprised that the lowering of XPD levels (an experiments that was presented in their previous study, Ma et al., 2018) is not shown here to fully challenge this hypothesis.

I was also wondering if da>CAK induces spindle modifications and triggers brain development defects: this important control is lacking in all figures. It is possible that excess of CDK1 activity may shorten the spindle due to CAK overexpression since tissue growth seems sensitive to the GAL drivers used, at least in disks. It is therefore difficult to interpret the data.

Minor points that nevertheless need to be amended.

1-Please measure the angles between centrosomes and the center of the nuclei (just before NEBD) to quantify the centrosome separation failure (similarly to figure S4).

2-Result section: avoid information that should be in the material and method section (ex: lane 183-185, also lane 950).

4-In the graphs it is sometimes difficult to see which samples are compared in the statistical tests (the blue bar stops between 2 samples). It is also not clear to me if the central brain volume, the Number of NBs per lobes, the OL volume is different between mms19 and mmsp19, da>CAK. It appears different to me but the P value is not shown.

5-This is a matter of taste but I feel the discussion is too long, does not go the point and distract the reader from the main message.

Other points.

6-I would prefer to see dot plot (+/-sd) instead of histograms or box plots. The exact n for each sample analyzed should be included in the figure legends.

7-The proteomic data are not needed. Why using a control that is also involved in MT dependent processes (lane 342)? Why not using GFP ? How the data can be interpreted ?

8-I am not sure that the neurite experiment can be interpreted because the MT binding properties of Mms19 have not been demonstrated in this study. It could be a secondary effect.

9-How is Aurora A kinase localized in mms19 mutant cells ?

10- Lane 440-443. I wouldn’t say that their previous studies have clearly shown mms19 interaction with MTs (Nag et al., 2018).

Reviewer #3: The article by Chippalkatti and colleagues entitled: “Mms19 promotes spindle microtubule assembly in neural stem cells through two distinct pathways” investigates the role of Mms19 in neuroblast cell division in the Drosophila central brain. They initially show that Mms19 mutant neuroblasts generate fewer GMCs than WT neuroblasts. They show that Mms19 mutant neuroblasts assemble spindles that are less robust than WT spindles and spend more time in mitosis. They further show that Mms19 contributes to microtubule stability and possible to bundling. The identification of Mms19 as a contributor of accurate mitosis is interesting and novel. The specificity of the phenotype in neuroblasts is also quite interesting. However, somehow at the end of the study we do not understand several observations and the various observations appear disconnected in order to understand the findings described here. I think the authors should address some important points to make this study a stronger candidate for Plos Genetics. Otherwise, the article appears rather descriptive and preliminary.

Major points:

1- Why neuroblasts are more sensitive to the lack of Mms19? In other words, what makes these cells so sensitive to the loss of this MAP, when compared to other brain cells from the optic lobe, for instance?

2- If neuroblast numbers are not that affected (Figure 1D), why the brain volume is decreased? Is the number of GMCs or neurons responsible for this? Also, when I look at the pictures shown in Figs 1A-B, it really seems that Mms19 mutant have fewer of these cells (Mira positive). The quantifications, however do not seem to show this. How do the authors explain this? I wonder if this relates with developmental stages. The extension of larval stages in the mutant might make a staged analysis more difficult, but maybe worth considering?

3- In Figure 2, I am not sure if understand what the authors mean by spindle orientation. Mis-orientation should be quantified towards a reference point, like a polarity crescent or the position of the daughter cells, which in this system should always be the same-see the Gonzalez lab papers.

4- Why is mitosis prolonged in the Mms19 mutant neuroblasts? Is it dependent on the spindle assembly checkpoint? Since nevertheless these cells seem to exit mitosis, do they generate aneuploid neuroblasts? Can this be at the basis of the decrease in brain size?

5- I am not sure that I agree with the interpretations based on the data shown in Figure 5. The authors show spindles in that Mms19 mutant neuroblasts and controls revealed by a-tubulin labeling. The spindles appears indeed very different, smaller and less organized. But the authors state that that Mms19 is required for spindle repolymerization. To me it does not seem a problem of microtubule re-polymerization. The microtubule mass is quite impressive in the mutant situation, just not well sorted into a bipolar array. But if I look at time t-30s, the array emanating from the centrosomes appears quite impressive in the mutant and comparable to controls. Maybe the problem is in establishing the length and not in generating microtubules after depolymerization?

6- I wonder how the authors build a model around their data related with TACC. TACC, has a minor phenotype in neuroblasts. Could it be that there is some redundancy between the two proteins? Did they make double mutants? Along the same lines, the table of interactors does not show TACC or any major MAP with important roles in spindle assembly. However some motors are present, which might explain the “abnormal spindle phenotypes”. Have they considered the option that maybe Mms19 plays a role in recruiting or activating a motor?

7- Several recent papers have analysed and compared mitotic spindles in different tissues or at different developmental stages. Maybe these should be included in either the introduction and or discussion?

**Have all data underlying the figures and results presented in the manuscript been provided?**

Reviewer #1: Yes

Reviewer #2: None

Reviewer #3: None

PLOS authors have the option to publish the peer review history of their article (what does this mean?). If published, this will include your full peer review and any attached files.

Reviewer #1: No

Reviewer #2: No

Reviewer #3: No

---

## [Decision Letter · Decision Letter 1]

30 Sep 2020

Dear Dr Suter,

Thank you very much for submitting your Research Article entitled 'Mms19 promotes spindle microtubule assembly in neural stem cells through two distinct pathways' to PLOS Genetics. Your manuscript was fully evaluated at the editorial level and by independent peer reviewers. Both reviewers found that your revised manuscript has clearly improved. Nevertheless, reviewer 1 made important comments, which should be addressed before we can offer publication of your manuscript. These comments, however, do not imply to make additional experiments. 

We therefore ask you to modify the manuscript according to the review recommendations before we can consider your manuscript for acceptance. Your revisions should address the specific points made by each reviewer.

[LINK]

Yours sincerely,

Jean-René Huynh

Associate Editor

PLOS Genetics

Gregory P. Copenhaver

Editor-in-Chief

PLOS Genetics

Reviewer's Responses to Questions

**Comments to the Authors:**

Reviewer #1: This is a well executed study with interesting results that are of relevance to the readership of Plos Genetics.

The manuscript has now greatly improved, but I feel the authors should clarify a few things before publication.

FIg1: Line: 182: "These observations indicated that the Mms19P

183 CB NBs probably did not proliferate enough to produce the normal amount of neuronal tissue [...]."

I still think that Fig 1 still shows that the CB is not so much affected, but that mitosis in the neuroepithelium is strongly reduced (PH3 signal/ Fig 1B the red area changes much less than the green). When you do live imaging on brains , it is obvious that the neuroepithelium and the neuroblasts it produces divide a lot. The authors should say that the brain size defects are more likely to stem from defects in highly proliferative areas like the neuroepithelium. If they want to insist that the slower CB neuroblast divisions are causing this, they need to do more experiments using Gal4 drivers that allow specifically testing CB neuroblast rescue in Mms19p mutants. It is a less elegant link to the subsequent experiments on neuroblasts. But those are still very justifiable and a very good system to understand mechanics that regulate spindle assembly.

Fig 2: I now realise that Fig 2G right panel is quite unlucky, is this the best picture showing spindle misalignment? I am to so sure it does. If they had a better one that would be desirable. As their interpretation that the degree of spindle misalignment is not in the range to cause neuroblast amplification (which is rather due to inheriting aPKC than Miranda symmetrically (Doe lab), perhaps rephrase that?). The interpretation/ conclusions here is nonetheless valid, i.e. spindle orientation defects are unlikely to explain it.

Fig 5:

line 331: "With this procedure, the NB spindles were completely depolymerized after incubation on ice for 30 min (Fig 5A)". Either add picture, or remove (Fig 5A) and say "not shown".

Reviewer #2: In this revised version, the authors have answered my questions and the new manuscript is much more focused on the important points. Moreover, clear efforts have been made on the figures.

I have a last minor points, (which will not require me to review a last this manuscript): It might be desirable to indicate that Aurora A also controls the degradation of cyclic B and add the corresponding reference (Caous et al., 2015), which may explain in part the increase in mitosis time.

**Have all data underlying the figures and results presented in the manuscript been provided?**

Reviewer #1: Yes

Reviewer #2: Yes

PLOS authors have the option to publish the peer review history of their article (what does this mean?). If published, this will include your full peer review and any attached files.

Reviewer #1: **Yes: **J Januschke

Reviewer #2: No

---

## [Editor Report · Decision Letter 2]

13 Oct 2020

Dear Dr Suter,

We are pleased to inform you that your manuscript entitled "Mms19 promotes spindle microtubule assembly in Drosophila neural stem cells" has been editorially accepted for publication in PLOS Genetics. Congratulations!

Yours sincerely,

Jean-René Huynh

Associate Editor

PLOS Genetics

Gregory P. Copenhaver

Editor-in-Chief

PLOS Genetics

Comments from the reviewers (if applicable):

**Data Deposition**

http://datadryad.org/submit?journalID=pgenetics&manu=PGENETICS-D-20-00873R2

**Press Queries**

---

## [Editor Report · Acceptance letter]

23 Oct 2020

PGENETICS-D-20-00873R2 

*Mms19* promotes spindle microtubule assembly in *Drosophila* neural stem cells 

Dear Dr Suter, 

We are pleased to inform you that your manuscript entitled "*Mms19* promotes spindle microtubule assembly in *Drosophila* neural stem cells" has been formally accepted for publication in PLOS Genetics! Your manuscript is now with our production department and you will be notified of the publication date in due course.

With kind regards,

Bailey Hanna

PLOS Genetics

On behalf of:
